

**Formation of Highly Absorptive Secondary Brown Carbon Through Nighttime**
**Multiphase Chemistry of Biomass Burning Emissions**
**Ye Kuang[1, *], Biao Luo[1], Shan Huang[1*], Junwen Liu[1], Weiwei Hu[2], Yuwen Peng[1], Duohong Chen[3],**
**Dingli Yue[3], Wanyun Xu[4], Bin Yuan[1], Min Shao[1]**
[1] Institute for Environmental and Climate Research, College of Environment and Climate, Jinan
University, Guangzhou, China.
[2] State Key Laboratory of Organic Geochemistry and Guangdong Key Laboratory of Environmental
Protection and Resources Utilization, Guangzhou Institute of Geochemistry, Chinese Academy of
Sciences, Guangzhou 510640, China
[3] Guangdong Ecological and Environmental Monitoring Center, State Environmental Protection Key
Laboratory of Regional Air Quality Monitoring, Guangzhou 510308, China
[4] State Key Laboratory of Severe Weather, Key Laboratory for Atmospheric Chemistry, Institute of
Atmospheric Composition, Chinese Academy of Meteorological Sciences, Beijing, China
Corresponding author: Ye Kuang (kuangye@jnu.edu.cn) and Shan Huang
(shanhuang_eci@jnu.edu.cn)



## Abstract

Biomass burning is a major global source of both primary brown carbon (BrC) and reactive trace gases in the atmosphere, thus exerts significant impacts on global climate and regional atmospheric chemistry. However, a substantial gap remains in our understanding of the nighttime evolution of biomass burning emissions. Here we present prominent nighttime formation of secondary organic aerosol (Night-OA) with strong absorptivity but markedly different spectral dependence from that of primary biomass burning organic aerosols, which was observed during autumn of the Pearl River Delta region of China when biomass burning plumes prevailed. Our results demonstrate that the formation of Night-OA appeared high dependence on both magnitudes of afternoon biomass burning emissions and available oxidants of $NO_2$ and $O_3$. Active nighttime $NO_3$ radical chemistry was characterized by quick $O_3$ depletion and almost zero concentration of NO, and the rapid decrease of $NO_2$ coincident with the quick nitrate formation suggests that the rapid $NO_2$ consumption supplied the $NO_3$ and $N_2O_5$ reaction chains. However, the quickest Night-OA formation occurred when nitrate formation ceased and relative humidity reached maximum, and mainly added mass to aerosol water abundant diameter ranges. This co-variation demonstrates that gas-phase and aqueous-phase chemistry of biomass burning precursors likely coordinated to promote the quick nighttime formation of Night-OA. Findings of this study highlight the nighttime darkening of biomass burning plumes through multiphase reactions and the proposed secondary BrC formation mechanisms may have broad implications in climate and air quality effects of biomass burning, such as the interaction between biomass burning plumes with water abundant pyroconvection cloud.



## 1 Introduction

Light absorbing organic aerosols termed as brown carbon (BrC) (Andreae and Gelencsér, 2006), absorbs solar radiation and warms the atmosphere, acts as potential photosensitizers and alters atmospheric oxidation capacity (Liu et al., 2020), thus impacting profoundly on atmospheric chemistry, air quality and climate (Jo et al., 2016). Biomass burning activities happen frequently across the globe due to natural and anthropogenic activities, injecting large amounts of fine organic aerosols (known as biomass burning organic aerosol, BBOA) and trace gases into the global atmosphere, thereby acting as a major global source of atmospheric primary organic aerosols and non-methane volatile organic compounds (Andreae and Merlet, 2001;Akagi et al., 2011;Andreae, 2019;van der Werf et al., 2017), thus exerting significant impacts on global climate (Liu et al., 2021;Yu et al., 2019;Yu et al., 2021). The freshly emitted BBOA is highly absorptive near ultraviolet wavelengths and considered as a major contributor to atmospheric BrC (Wang et al., 2016). A number of laboratory studies revealed that secondary organic aerosol (SOA) formed from the oxidation of biomass burning precursors (BBSOA) is also absorptive (Saleh et al., 2013), however, the importance of secondary BrC formed from biomass burning precursors remains uncertain.

Both daytime and nighttime chemistry play significant roles in aging biomass burning plumes and associated secondary SOA and BrC formation. Daytime aging of biomass burning emissions and its impacts on SOA formations have previously been extensively investigated (Hodshire et al., 2019) , which found that the formed SOA contributed substantially to BrC (Kumar et al., 2018;Saleh, 2020) and confirmed by field measurements. For example, Palm et al. (2020) observed that daytime oxidation of emitted phenolic compounds contributed a majority to BBSOA formation, with products being highly absorptive. Compared with daytime aging experiments of biomass burning emissions, laboratory studies representative of night chemistry are scarce. At night, the consumptions of $O_3$ and OH is much faster than their formation rates due to the absence of photochemical reactions, thus atmospheric concentration of these two oxidants reduce rapidly after sunset. The depletion of $O_3$ through $NO_2$ oxidation generates $NO_3$ radical that act as a major oxidant during nighttime periods and drives the tropospheric nighttime chemistry. Decker et al. (2019) investigated the nighttime chemical transformation in biomass burning plumes using a box model, which was initialized by aircraft observations, with results demonstrating $NO_3$ radical loss mostly due to reactions with biomass





burning volatile organic compounds (VOCs). In recent years, a number of laboratory studies using
$NO_3$ as oxidant were carried out to explore influences of nighttime aging on biomass burning emissions
related SOA formations and associated BrC evolutions (Cheng et al., 2020;Li et al., 2020a;Jiang et al.,
2019;Tiitta et al., 2016;Hartikainen et al., 2018). Their results demonstrated that both nighttime aging
of BBOA aerosols and biomass burning VOC precursors such as pyrrole can potentially act as
important sources of SOA and BrC. However, contributions derived from laboratories studies cannot
be directly linked with SOA and BrC contribution magnitudes within ambient nighttime atmosphere.
Kodros et al. (2020) further highlighted the importance of nighttime processing of biomass burning
emissions as an important global source of SOA based on combined results from laboratory and field
observations. Nevertheless, field measurements that observed nighttime evolutions of biomass burning
plumes and directly confirmed significant contributions of SOA and BrC from nighttime aging of
biomass burning emissions are highly in lack. Only one field measurement study has observed
substantial increase of BrC light absorption during a night-long biomass burning event and identified
nitroaromatics in abundance within aged BBOA aerosols, contributing greatly to light absorption (Lin
et al., 2017;Bluvshtein et al., 2017), based on which it was hypothesized that nighttime chemistry
involving $NO_3$ radical oxidation of primary BBOA might play significant roles in BrC transformation.
Nighttime conditions are typically characterized by high atmospheric relative humidity (RH) caused
by temperature decreases, therefore likely result in abundant aerosol water, which might favor aqueous
SOA and BrC formation (Wang et al., 2019b). However, most of previous laboratory studies have not
investigated the role of RH or aqueous phase chemistry in nighttime aging of biomass burning plumes.
Kodros et al. (2022) found that nighttime oxidation of biomass burning emissions were sensitive to
RH, however, could not conclude what roles RH was. Their results demonstrated that homogenous
gas-phase oxidation and subsequent condensation of lower-volatility vapors was probably the
dominant process, however, they could not rule out the possible role of heterogeneous oxidation
processes. Therefore, how nighttime $NO_3$ radical chemistry coordinates with aerosol aqueous or
heterogenous reactions under high nighttime RH conditions to affect SOA and BrC formations remains
unexplored, which is a substantial knowledge gap in the research field of nighttime chemical
transformation of biomass burning emissions and its role in SOA and secondary BrC formations.

In this study, we report substantial amounts of highly absorptive SOA likely formed during

nighttime mainly from biomass burning emissions. The potential formation mechanisms of nighttime





SOA formation are investigated based on real-time measurements of parameters such as gaseous
pollutants, aerosol physical and chemical properties and meteorological factors. Our results revealed
that coordinated nighttime multiphase chemistry of biomass burning emissions likely formed highly
absorptive SOA, which improved our current understanding on nighttime aging of biomass burning
emissions and might also have significant implications for cloud processing of biomass burning.
Additionally, this study is a companion paper to Luo et al. (2022), where we proposed an improved
absorption Ångström exponent (AAE) ratio method for deriving multiwavelength BrC absorptions
from multiwavelength aerosol absorption measurements using an aethalometer, and investigated
comprehensively size distribution, absorption and scattering as well refractive index of primary BBOA.

**2 Materials and Methods**

**2.1 Field measurements**

A field campaign was conducted from 30 September to 17 November 2019 at a regional
background site of the Peral River Delta region. This site locates at the country side of Heshan county,
about 55 km away from the megacity Guangzhou and at the top of hill with surroundings are small
villages and residential towns. Routine observations of air pollutants such as carbon monoxide, ozone
nitrogen dioxides and $PM_{2.5}$ (particulate matter with aerodynamic diameter less than 2.5 $\mu$m), and
meteorological parameters such as air relative humidity (RH), temperature, wind speeds and directions
were managed by the provincial environmental monitoring authority and this site is authorized as a
regional background supersite. During the observation period, intensive aerosol measurements
including aerosol optical properties, aerosol size distributions as well as aerosol chemical compositions
were also performed to investigate relationships between aerosol physical properties and aerosol
chemical compositions. The aerosol scattering properties and aerosol hygroscopicity were measured
using a humidified nephelometer (Aurora 3000) system (Kuang et al., 2021). Aerosol absorptions of
multiple wavelengths were measured using an aethalometer (Magee AE33, (Drinovec et al., 2015)).
Aerosol size distributions were measured jointly by using a scanning mobility particle sizer (SMPS,
TSI 3080) and an aerodynamic particle sizer (APS; TSI Inc., Model 3321). More details on the site
and set-up of instruments please refer to Kuang et al. (2021) and Luo et al. (2022).
The submicron aerosol chemical compositions were measured using a soot particle high-
resolution time of flight aerosol mass spectrometer (SP-AMS, Aerodyne Research, Inc., Billerica, MA,



USA). The set-up and validation of SP-AMS measurements were performed and discussed in Kuang,
et al. (Kuang et al., 2021), thus not detailed here. Source identification of organic aerosols was
performed using the commonly used positive matrix factorization (PMF), two primary OA factors and
four secondary OA factors are identified, and the determination of PMF factors are thoroughly
discussed in Luo et al. (2022). The two primary OA factors include biomass burning organic aerosols
(BBOA) and a hydrocarbon-like organic aerosols (HOA). The biomass burning emissions represent
the most important primary sources during the observations as discussed in Luo et al. (2022), and its
most prominent activities were usually observed near sunset (Fig.1a).The four SOA factors including
more oxygenated organic aerosols (MOOA, O/C=1), less oxygenated organic aerosols (LOOA,
O/C=0.72), nighttime-formed organic aerosols (Night-OA, O/C=0.32) and aged BBOA (aBBOA,
O/C=0.39). The Night-OA factor was characterized by its obvious correlation with nitrate and they
both exhibited obvious increase after sunset (Fig.1a). The name of aBBOA was originally because of
its correlation with $C_6H_2NO_4^+$ which is a typical fragment of the aged BBOA component nitrocatechol
(Bertrand et al., 2018). The mass spectral profiles and time series of these organic aerosol factors were
shown in supplement of Luo et al. (2022), and details about the determination of these factors are
introduced in supplement of Luo et al. (2022). Note that the AE33 measurements were only available
until 1th of November, while SP-AMS measurements were available until 18th of November, resulting
in different time frames of different timeseries.
Given that PMF analysis is fundamental to our study, the mass spectral profiles of factors are
provided in Fig. S1, and key aspects of the resolved results are explained here, particularly concerning
the O/C characteristics and the naming of aBBOA and Night-OA. In previous studies (Kuang et al.,
2021;Luo et al., 2022), we already realized that the correlation between aBBOA and $C_6H_2NO_4^+$ was
actually weak (R=0.31), suggesting that it might not fully be constituted of aging products of primary
BBOA considering its O/C was even lower than that of BBOA. However, aBBOA exhibited similar
diurnal behavior to LOOA showing clear daytime photochemical production plus an evening peak
around 19:00 (local time) just after the peak hour of BBOA as shown in Fig.S1. Also, the evening peak
value of aBBOA between 19:00 and 22:00 of each day was found to be moderately correlated with the
noon peak in the next day (R =0.55; (Wu et al., 2024)). This tells that aBBOA could be emitted from
biomass burning. However, it could be speculated that only a small portion (aBBOA accounts for an
average of 8% of the mass increase during identified biomass burning events, Fig. 3 of Luo et al.



(2022)) is directly emitted. Most of it likely originated from the gas-phase oxidation of biomass
burning emitted VOC precursors considering that aBBOA mainly added mass to diameters of
condensation mode (see discussions about size distributions of OA factors in supplement of Luo et al.
(2022)), which could have a low O/C during short oxidation periods, as demonstrated in previous
indoor experiments on biomass burning emissions (Yee et al., 2013; Ahern et al., 2019). This
hypothesis is further supported by the fact that increases in aBBOA loading enhance organic aerosol
hygroscopicity despite its low O/C, as demonstrated by Kuang et al. (2021), whereas primary organic
aerosols have not been observed to enhance overall organic aerosol hygroscopicity (Kuang et al., 2020).

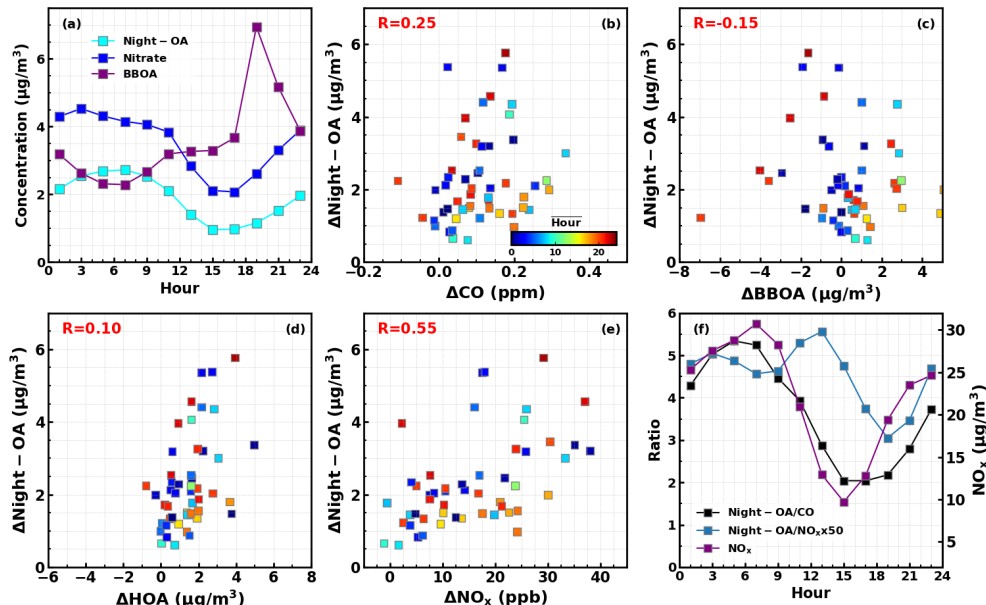

**Figure 1. (a)** Diurnal variations of nitrate, Night-OA and BBOA; **(b-e)** Relations between increases of Night-OA and increases of CO, BBOA, HOA and NOx for identified Night-OA increase cases, colors of scatter plots represent the average time of Night-OA increase cases; **(f)** Diurnal variations of ratios Night-OA/CO, Night-OA/NOx in the left axis and NOx in the right axis.

The Night-OA factor has a relatively low O/C ratio of 0.32, raising the question of whether it

originates from primary emissions or secondary formation. As discussed in Luo et al. (2022), traffic,
cooking (The HOA and COA and were not separated in the PMF results although the hydrocarbon
factor was named HOA as discussed in Kuang et al. (2021)), and biomass burning are likely the
dominant primary sources during this campaign. If Night-OA were a primary source, it would be
expected to increase alongside other primary sources. We identified most Night-OA increase events



and examined their correlation with variations in other primary sources, as shown in Fig. 1b-e. This
analysis reveals that Night-OA increases were typically observed after sunset, though occasionally
during the daytime. Night-OA increases showed weak correlations with changes of CO (R=0.25), HOA
(R=0.1), and BBOA (R=-0.15), but a moderate correlation with NOx (R=0.55). During significant
biomass burning events (indicated by substantial BBOA increases), the concentration of Night-OA
actually decreased on average (Fig. 3 of Luo et al. (2022)), suggesting that Night-OA is unlikely to be
emitted from biomass burning. We also identified all significant HOA increase events that did not
coincide with biomass burning and analyzed the average HOA increase and variations in other aerosol
components (Fig. S2). It shows that, despite significant HOA increases, the average mass concentration
of Night-OA remained almost unchanged, indicating that Night-OA is also unlikely to originate from
HOA-associated emissions. Therefore, the weak but positive correlations between Night-OA and HOA
as well as CO are likely associated with the accumulation characteristics of primary emissions after
sunset. The higher correlations between Night-OA and NOx may also result from the accumulation of
NOx starting in the afternoon when photochemical depletion is weaker. Another possibility to consider
is whether Night-OA increases could be associated with plumes containing higher NOx transported
from other regions. We investigated the diurnal variations of the Night-OA/CO and Night-OA/NOx
ratios, observing persistent increases in Night-OA/CO and Night-OA/NOx ratios when significant
Night-OA formation began. This suggests that Night-OA is likely formed through secondary processes,
consistent with that it was also correlated with nitrate (R=0.67). The low O/C of Night-OA, still higher
than that of the primary factor HOA, was determined by a high amount of $C_xH_y^+$ ions in spite of
significant intensity of oxidation tracers $C_2H_3O^+$ and $CO_2^+$, suggesting that Night-OA was oxidation
products with low oxidation state during the nighttime. Similar situation was previously found in study
at Bakersfield of USA in which Liu et al. (2012) identified SOA factors as alkane-SOA and aromatic-
SOA with moderate O/C (0.27-0.36).  Meanwhile, NOx potentially promoted its formation, given the
highest N/C ratio of Night-OA among all resolved factors. This will be discussed further in Sect 3.2.

In summary, both aBBOA and Night-OA are not likely primary, while the naming of aBBOA and

Night-OA factors might not be perfect, we retain these terms in this study for consistency with our
previous work (Kuang et al., 2021;Luo et al., 2022;Wu et al., 2024).
**2.2 Quantification of BrC absorptions**

In Luo et al. (2022), we proposed an improved AAE ratio method to subtract brown carbon (BrC)



absorptions from measured total aerosol absorptions during this field campaign. Therefore, the details
about discussions of this method please refer to Luo et al. (2022), and we only introduce briefly the
philosophy of the new method here. The essential part of deconvolving BrC absorptions from total
aerosol absorptions is the adequate representation of black carbon (BC) spectral absorptions using the
Ångström exponent law. Results of previous studies (Wang et al., 2018a;Li et al., 2019a) demonstrated
that significant wavelength dependence of $AAE_{BC}$ and constant assumption of $AAE_{BC}$ in BrC
absorption retrievals might lead to significant bias. Thus, the AAE ratio defined as
$R_{AAE}(\lambda)=AAE_{BC,\lambda-880}/AAE_{BC,950-880}$ was proposed to tackle spectral $AAE_{BC}$ variations, and on-line
measurement data of $AAE_{950-880}$ were used as $AAE_{BC,950-880}$ due to negligible absorption
contributions of BrC at wavelengths of 880 nm and 950 nm. Thus, BrC absorptions ($\sigma_{BrC}$) at different
wavelength ($\lambda$) including 370 nm, 470 nm, 530 nm, 590 nm and 660 nm can be derived using the
following formula, where $\sigma_a$ is measured aerosol absorption:

$$\sigma_{BrC}(\lambda) = \sigma_a(\lambda) - \sigma_{BC}(880\ nm) \times \left(\frac{880}{\lambda}\right)^{AAE_{BC,950-880} \times R_{AAE}(\lambda)} \qquad (1)$$

As the sophisticated discussions presented in Luo et al. (2022), variations of many factors such
as BC refractive index, coating shell refractive index as well as BC mixing state, and BC mass size
distributions (Li et al., 2019b) might influences the magnitudes of $R_{AAE}(\lambda)$. However, sensitivity tests
shown in Luo et al. (2022) clearly concluded that BC mass size distributions dominated $R_{AAE}(\lambda)$
variations. In Heshan campaign, the BC calibration for SP-AMS was not available, so, the carbon
signals were used to retrieve shape of BC mass size distribution, and the total mass was constrained
by the ratio between integrated $C_x$ ($C_1$-$C_9$) signals and BC mass concentrations provided by the AE33.
The real-time measured carbon fragments ($C_x$) distributions by the SP-AMS were therefore used to
distribute the total BC mass to different diameter bins to calculate $R_{AAE}(\lambda)$ as introduced in Luo et al.
(2022). Details of the calculation can be found in its supplement. The average and standard deviations
of $R_{AAE}$ (370), $R_{AAE}$ (470), $R_{AAE}$ (520), $R_{AAE}$ (590) and $R_{AAE}$ (660) are 0.79(±0.044), 0.85(±0.038),
0.88(±0.035), 0.9(±0.035) and 0.93(±0.031) during the observation period.
**3 Results and discussions**
**3.1 Highly absorptive SOA formed during nighttime.**





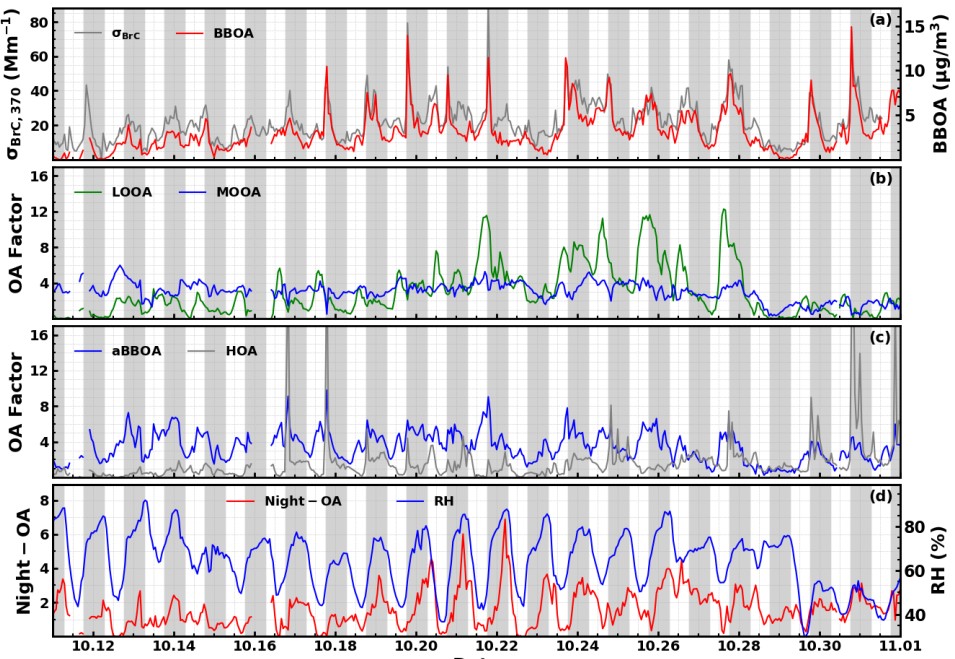

**Figure 2**. Timeseries of **(a)** brown carbon absorption at 370 nm ($\sigma_{BrC,370}$) in the left axis and BBOA in the right axis; **(b)** LOOA and MOOA; **(c)** aBBOA and HOA; **(d)** Night-OA in the left axis and RH in the right axis. Shaded areas represent nighttime periods.

As reported by Luo et al. (2022), biomass burning events happened frequently during the
observation period and biomass burning was the dominant source of primary organic aerosols with the
average BBOA/HOA ratio of 3.3. The dominant contribution of biomass burning to primary aerosol
emissions together with the strong biomass burning emissions before dusk as well as sometimes
daytime emission characteristics (Luo et al., 2022) provided a unique opportunity to explore the
nighttime chemistry associated with biomass burning and its impacts on atmospheric BrC evolution in
biomass burning plumes. Timeseries of retrieved $\sigma_{BrC,370}$ is shown in Fig.2a. It shows that BBOA
varies quite consistently with $\sigma_{BrC,370}$ and they are highly correlated (R=0.88), demonstrating the
dominant contribution of BBOA to BrC absorptions. However, the correlation coefficient between
BBOA and $\sigma_{BrC}$ decreases as the wavelength increases, i.e., the correlation coefficients between $\sigma_{BrC}$
at 470 nm, 520 nm, 590 nm, 660 nm and BBOA are 0.83, 0.8, 0.76, 0.69. In addition, as shown in
Fig.2a, coordinal variations between BBOA $\sigma_{BrC,370}$ are usually seen during daytime especially during
the dusk BBOA spike periods, however, the $\sigma_{BrC,370}$ usually deviates substantially from BBOA



variations during the nighttime (gray areas in Fig.2). The average diurnal variations of both $\sigma_{BrC,370}$
and the ratio $\sigma_{BrC,370}$/BBOA is presented in Fig.3a, and quick $\sigma_{BrC,370}$/BBOA increase were observed
during nighttime before 06:00 LT. These results demonstrate that organic aerosol components other
than BBOA also contribute substantially to BrC absorption and differ much at different wavelengths.

The time series of OA factors other than BBOA are also shown in Fig.2b-d. As analyzed in Kuang

et al. (2021), SOA contribute dominantly to total OA mass (SOA mass fraction>70% on average)
during this field campaign. Most prominent features of SOA formations are the quick daytime
formation of LOOA, aBBOA and nighttime formation of Night-OA, while MOOA exhibit almost no
diurnal variations and are mostly associated with the regional airmass. The average mass absorption
efficiencies (MAEs) (m$^2$/g) of different OA factors are retrieved using multivariate linear regression
method (Fig.S3), and the deduced average MAEs values at 370 nm for BBOA, aBBOA, HOA, LOOA,
MOOA and Night-OA are 3.8, 0.84, 0.24, 0, 1, 2.3 m$^2$/g respectively. Note that negative value of about
-0.1 is retrieved if LOOA values were inputs of the multivariate linear regressions, demonstrating quite
low absorptivity of LOOA, thus MAE of LOOA is treated as zero. As shown in Fig.2b, rapid LOOA
formation episodes happened frequently during daytime but $\sigma_{BrC,370}$ still varies only with BBOA
during that period, which confirms the white property of LOOA. The derived MAE of HOA is very
small which is consistent with the low absorptivity of HOA during HOA spikes. The prominent BrC
factors are identified as BBOA, Night-OA, MOOA and aBBOA. The most surprising part is the highly
absorptive property of Night-OA whose absorptivity is only lower than that of BBOA but much higher
than other OA factors. Based on derived MAE values for OA factors as well as their mass
concentrations, contributions of OA factors to $\sigma_{BrC,370}$ are estimated and their diurnal variations are
shown in Fig.3b. It reveals that Night-OA accounts for the second largest contribution to BrC
absorption during nighttime and even reaches beyond the contribution of BBOA near local time 06:00
(>30%), which explains the observed substantial nighttime deviation of $\sigma_{BrC,370}$ from BBOA as shown
in Fig.3a. The average contributions of BBOA, Night-OA, MOOA and aBBOA to $\sigma_{BrC,370}$ are 50%,
20%, 16%, 12%, respectively. The time series of estimated OA contributions to BrC absorption at 370
nm are shown in Fig.S4, it tells that Night-OA sometimes contribute dominantly (>50%) to $\sigma_{BrC,370}$
especially when rapid increase of Night-OA appeared due to nighttime secondary formation.

Different BrC components usually exhibit different absorption spectral dependences (Laskin et

al., 2015). Diurnal variations of AAE$_{370-470, \text{ BrC}}$ are investigated and shown in Fig.3c. The AAE$_{370-470,}$

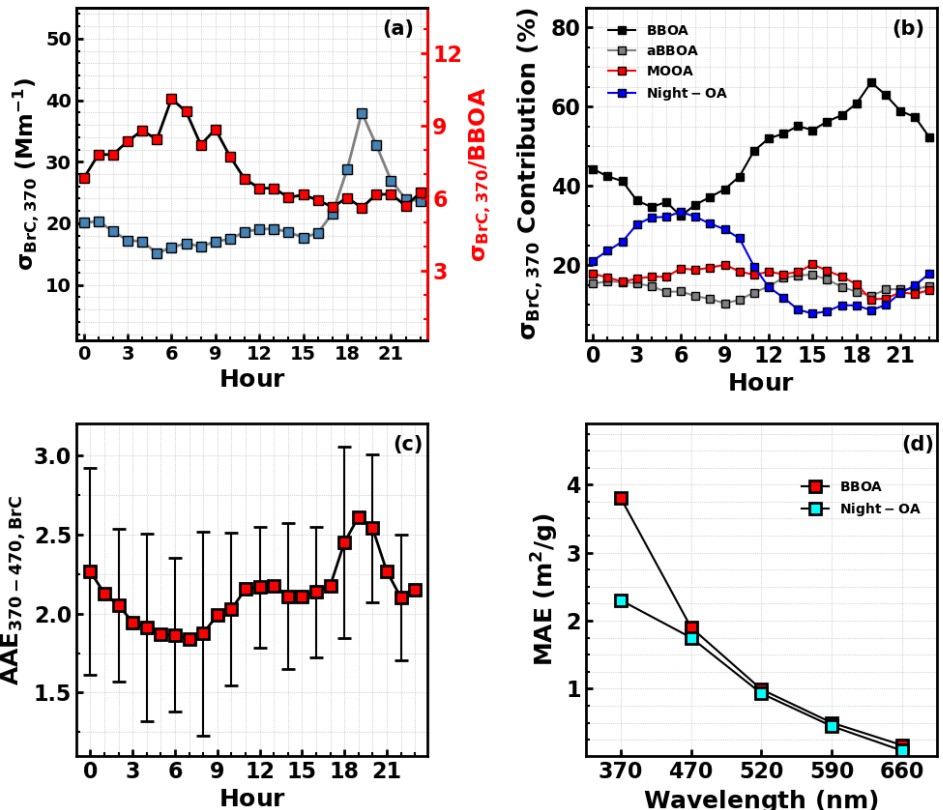

**Figure 3. (a)** Diurnal variations of $\sigma_{BrC,370}$ and the ratio $\sigma_{BrC,370}$/BBOA; **(b)** Diurnal variations of contributions of different OA factors to $\sigma_{BrC,370}$; **(c)** Average diurnal variations of brown carbon absorption angstrom exponent between 370 nm and 470 nm; **(d)** Retrieved mass absorption efficiencies (MAE, ) of BBOA and Night-OA at multi-wavelengths.

$_{BrC}$ exhibits distinct diurnal variations, with one peak at dusk when BBOA reaches its highest mass
concentration and a trough at the time when Night-OA contributes most to and BBOA contribute the
least to $\sigma_{BrC,370}$. This phenomenon suggests that spectral dependence of Night-OA absorptivity differs
much with that of BBOA. The multivariate linear regression method is thus also used for retrieving
MAEs of BBOA and Night-OA at wavelengths of 470 nm, 520 nm, 590 nm and 660 nm (the
performance of using retrieved MAE values at multiple wavelengths are shown in Fig.S3), and the
retrieved spectral dependence of BBOA and Night-OA absorptivity are shown in Fig.3d. The results
show that Night-OA absorbs as strong as that of BBOA at visible wavelength ranges, highlighting a
more important role of Night-OA in BrC absorption than expected from the retrieval of $\sigma_{BrC,370}$. The



retrieved $AAE_{370-470}$, $AAE_{470-590}$ for BBOA and Night-OA are 3 and 5.9, 1.3 and 6, respectively, which
explains the observed quick decrease of $AAE_{370-470, BrC}$ during nighttime. The direct quantification of
$AAE_{370-470}$, $AAE_{470-590}$ for BBOA is difficult due to the entanglement of BC absorption and thus rarely
reported. The retrieved $AAE_{470-590}$ for BBOA and Night OA are in general consistent with the AAE of
bulk BrC solutions extracted using different solvents which were sampled during and after a nighttime
nationwide biomass burning event (Lin et al., 2017).

**3.2 Precursors and Possible Mechanisms of the Night-OA formation.**

As shown in Fig.1a and Fig.2d, the Night-OA concentration increased during the nighttime, while
usually decreased and reached near zero in the afternoon, so the Night-OA factor is characterized by
its rapid nighttime formation and quick daytime evaporation. In general, SOA can either be formed
through condensation of gas-phase chemically aged low- or semi-volatile VOC precursors following
the gas-particle partitioning theory or formed in the aqueous phase through further oxidation of water-
soluble primary VOCs as well as secondary products of gas-phase VOC aging processes, with the
former referred to as gasSOA and the latter referred to as aqSOA. Several recent researches reveal that
SOA can also be formed through oxidation of semi-volatile components evaporated from emitted
primary organic aerosols in gas phase (Palm et al., 2020) or in the aqueous phase (Wang et al., 2021).
The sources and formation pathways of Night-OA is of great concern in meriting the importance of
Night-OA formations in global atmosphere and paving the way for designing targeted laboratory
experiments in future.
The VOC profiles of this observation site are quite complex as mixtures of both different
anthropogenic sources and natural sources  (Song et al., 2019). However, prominent and continuous
Night-OA formation events observed from 19 to 22 October which are accompanied with frequently
observed sharp increase of BBOA mass concentrations before the fall of night bring hints that Night-
OA formation might be associated with biomass burning emissions. Indeed, the air is always moving
and to be lagrangian, the air after midnight might differ with those before midnight. However, the
relations between Night-OA signals after midnight and emission signals before midnight might still
deliver some clues.  The average nighttime mass concentrations of Night-OA from 22:00 LT to 06:00
LT (the next day) are plotted against the average BBOA mass concentrations from 16:00 LT to 22:00
LT of this campaign and the results are shown in Fig.4a. It shows that Night-OA formation is positively
correlated with BBOA before the night which supports the speculation that Night-OA formation is a



result of the nighttime chemistry of biomass burning emissions. The nighttime oxidant levels indicated
by the concentrations of Ox (NO$_2$+O$_3$) represented by colors in Fig.4a demonstrates that the highest
Night-OA formation was accompanied with strongest biomass burning emissions and most abundant
oxidants, suggesting that the importance of nighttime aging processes in Night-OA formation.
Following evidences show that gas-phase and aqueous-phase chemistry of biomass burning precursors
likely have coordinated to promote the quick nighttime formation of Night-OA.

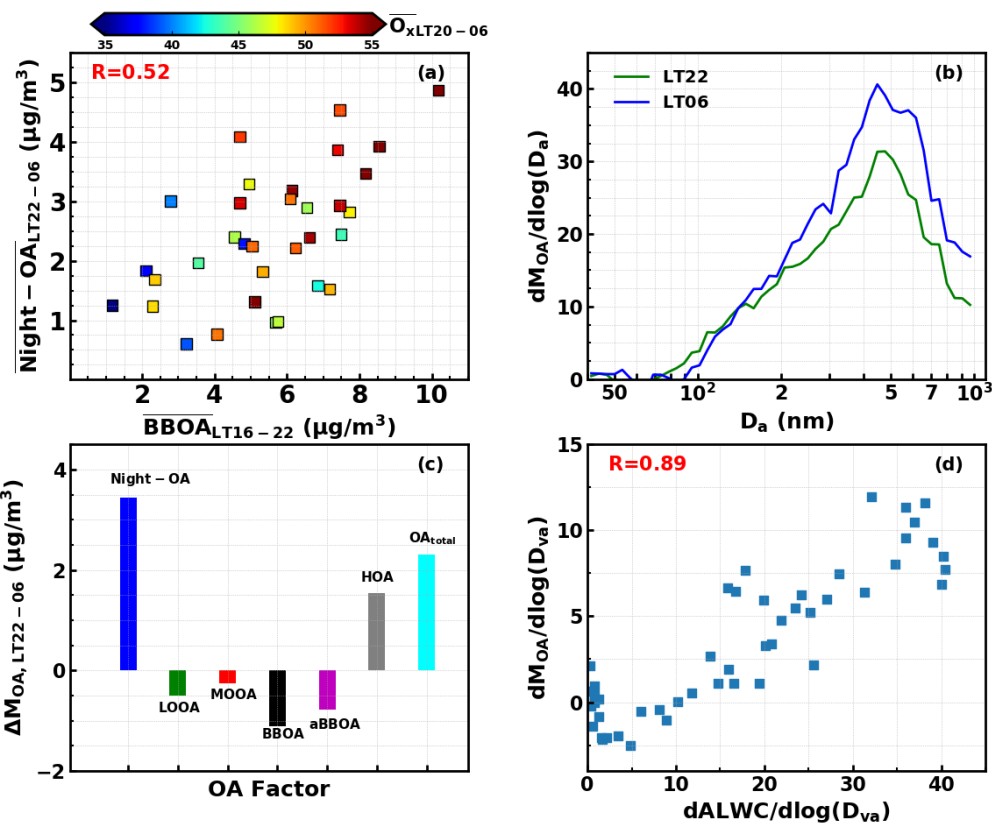

**Figure 4.** **(a)** Correlations between average nighttime Night-OA mass concentrations and average BBOA mass concentrations over LT16-22, colors corresponding to the average Ox (NO$_2$+O$_3$) mixing ratio (ppb) during the night; **(b)** Average evolutions of organic aerosol mass size distributions from local time 22:00 (20, 21 October ) to 06:00 (21, 22 October); (c) Average differences for organic aerosol (OA) mass concentration corresponding to (b); **(d)** The correlations between increase of size-resolved organic mass concentrations in (c) (LT06 minus LT22) with average size-resolved aerosol liquid water content (ALWC).

Three obvious Night-OA formation episodes which lasted more than three days, as shown in

Fig.S5, were observed during the entire field campaign. The peak Night-OA mass concentration of





each night increased during these episodes and were accompanied with the increase of nighttime peak
RH during each episode, suggesting that abundant nighttime aerosol water might play significant roles
in Night-OA formation. More solid clues could be found from different behaviors of gasSOA and
aqSOA formations in modifying the aerosol size distribution. The gasSOA forms through condensation
following partitioning theory thus adding mass mainly to condensation mode which contributes most
to aerosol surface area concentrations. Whereas, aqSOA formation depends on amounts of liquid water
content thus adding mass mainly to the mode where most aerosol water resides. The evolution of
average OA size distribution during the night of 20 and 21 October when most prominent Night-OA
formation occurred is illustrated in Fig.4b. During the two nights from 22:00 LT to 06:00 LT, Night-
OA formations contributed most to the mass concentration increase of the entire OA and HOA
contributed less, while mass concentrations of all other OA factors have decreased and partially
balanced out the OA increase (Fig.4c). HOA emissions mainly added mass to diameter ranges of 100-
300 nm as demonstrated by Luo et al. (2022). However, as shown in Fig.4b, substantial increase of
mass concentrations at diameter range of > 300 nm occurred even all other SOA factors showed a
decreasing trend for the case shown in Fig.4c, suggesting the substantial OA mass increase of larger
than 300 nm are contributed by the Night-OA increase. The results shown in Fig.4d showed that the
size-resolved increase of OA mass for abovementioned cases (Fig.4b and c) was highly correlated with
size-resolved aerosol liquid water content (details about the size resolved aerosol water content
calculation are presented in Sect 1.2 of the supplement), demonstrating that the Night-OA formation
added mass mainly to aerosol water abundant diameter ranges. The average RH of the cases shown in
Fig.4b-d is around 80%, with a corresponding average hygroscopicity parameter $\kappa$ of 0.26 measured
by the humidified nephelometer system, demonstrating a growth factor of ~ 1.27 (water thickness of
~81 nm for aerosol diameter of about 300 nm), thus not a thin film of water for heterogeneous reactions,
but more likely formed through dark aqueous reactions.

In addition, as shown in Fig.S6, the Night-OA decreased quickly during daytime, which is beyond

the dilution effect of boundary layer development (indicated by rapid decrease of Night-OA/CO as
shown in Fig.S6) thus implying the substantial daytime loss of Night-OA which might be caused by
several processes, such as partitioning and photodegradation (Wang et al., 2023). The partitioning
dynamics of gasSOA and aqSOA also differ much, which might add more information in looking into
the Night-OA formation pathway. Following the rule of partitioning theory, evaporation equilibrium





of the condensation of gasSOA is tightly associated with air temperature. Equilibrium of the reversible
aqSOA formation greatly depends on aerosol water changes, while evaporation of irreversible SOA
production through aqueous pathways are quite sophisticated (Tong et al., 2021). It was found that the
percentile of Night-OA mass loss during the daytime (07:00 to 16:00 LT) are more tightly correlated
with RH decrease compared with temperature increase (Fig.S7). HOA also exhibits obvious
evaporation during daytime (indicated by rapid decrease of HOA/CO Fig.S6), however, the daytime
loss of HOA seems more tightly correlated with temperature increase as shown in Fig.S8. The
phenomenon that Night-OA daytime loss is more correlated with RH decrease implied that Night-OA
possibly co-evaporated with water vapor as RH decrease. This might serve weakly but still another
supporting clue for that Night-OA were likely formed through aqueous pathways and maybe reversible.

As revealed by Fig.4a that nighttime oxidation levels play significant roles in Night-OA formation,

the question leaves what's the role of nighttime gas-phase chemistry in promoting the Night-OA
formation.

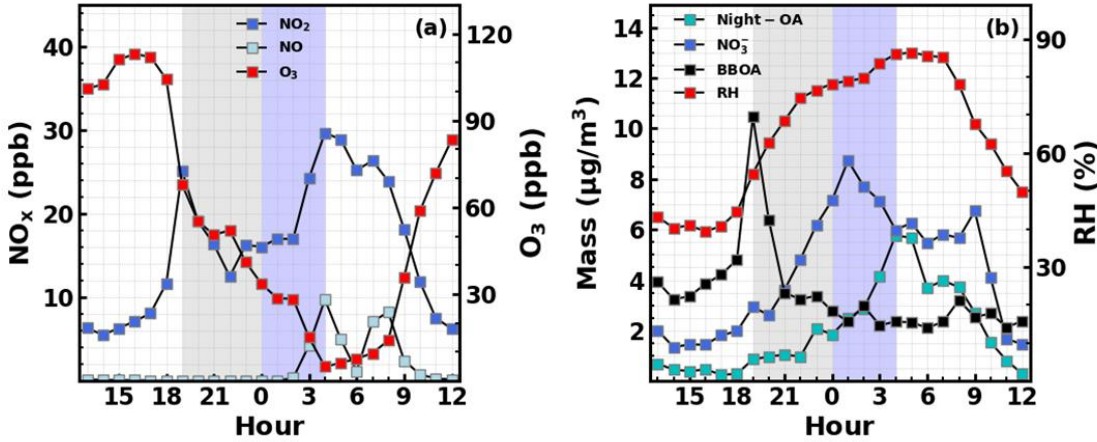

**Figure 5**. Average diurnal variations of **(a)** NO$_2$, NO, O$_3$; **(b)**Night-OA, nitrate, BBOA and RH during the two nights: 20 to 21 October when most prominent Night-OA formation occurred. Shaded areas represent periods of Night-OA increased, and blue parts correspond to remarkable increase period of Night-OA.

The average diurnal variations of gas pollutants and meteorological parameters typical of

nighttime chemistry including NO, NO$_2$, O$_3$, Night-OA, nitrate, RH as well as BBOA for the most
prominent Night-OA formation case (20 to 21, October, consistent with Fig.4b) are shown in Fig.5. At
night, NO$_2$ and NO$_3$ radical do not photolyze, NO reacts rapidly with O$_3$ and as a result almost all NOx





is concerted to $NO_2$ (Fig.5a). $NO_2$ will further react with $O_3$ to form $NO_3$ radical which is the most
important gas-phase oxidant during the nighttime (Chapleski et al., 2016). In this case, $NO_2$
concentration increased substantially before the sunset but the trend ended and shifted suddenly to
decrease when BBOA emissions were highest and then decreased rapidly along with the dropping
BBOA mass concentration. The reduction of BBOA mass concentration mainly attributed to the end
of local combustion events might also be associated with BBOA evaporation after its strong emissions
due to the mix of biomass burning plumes with background airmass, and this dilution effects shall
deliver semi-volatile VOCs such as phenolic compounds  from particle phase to gas phase (Palm et al.,
2020). The Night-OA increased slowly during the rapid $NO_2$ decrease phase but nitrate concentration
increased substantially with decreasing $O_3$ concentration (still higher than 30 ppb as shown in gray
shaded areas). The hydrolysis of $N_2O_5$, which is formed from $NO_2$ addition of $NO_3$ radicals, represents
an important pathway of nitrate formation during nighttime period. The rapid decrease of $NO_2$
coincident with the quick nitrate formation implies that the rapid $NO_2$ consumption supplied the $NO_3$
and $N_2O_5$ reaction chains, providing abundant $NO_3$ radical during the initial stage of Night-OA
formation. The active nighttime $NO_3$ chemistry and its impacts on nitrate formations during the
observation periods were further confirmed by Yang et al. (2022) who conducted box model
simulations. In addition, Li, et al. (2020a) demonstrated that night $NO_3$ radical darkened the BBOA
with the MAE enhancement ratio range from 1.3 to 3.2 for optical wavelength of less than 650 nm.
The retrieved average MAE of BBOA through multilinear fitting was higher than the average MAE of
freshly emitted BBOA during BBOA spikes as reported by Luo et al. (2022) (3.8 vs 2.6 m$^2$/g),
suggesting the darkening of primary BBOA which is consistent with the prevailing nighttime $NO_3$
chemistry processes during the observations. The highest Night-OA production rate occurred when
both $NO_2$ and NO began to increase ($O_3$ still decreased rapidly and reached below 30 ppb, shown as
the blue shaded area) and the RH reached near the its maximum, which further highlights the crucial
role of aerosol liquid water content in Night-OA formation. However, the quick increase of $NO_2$ and
NO implies that the dominant contribution of $NO_2$ formation to $O_3$ depletion, and the $NO_3$ radical
chemistry have ceased and likely did not directly participate in the succeeding quick aqueous-phase
Night-OA formation. Nevertheless, quick depletion of $O_3$ and increase of $NO_2$ with the quickest Night-
OA formation occurred demonstrates that the $NO_2$ might play significant roles in the subsequent quick
Night-OA formation. These results demonstrate that the aqueous-phase processing of biomass burning



emissions with abundant $NO_2$ could likely form highly absorptive SOA, while the nighttime gas-phase
chemistry with typically high $NO_2$, $NO_3$ radical and RH could likely magnify the Night-OA formation.
A previous field study (Lin et al., 2017) demonstrated that most chromophores were nitroaromatic
compounds (NAC) during the observed nighttime bonfire event with the major contribution to the
solvent extractable BrC, and the NAC contribution to BrC absorption increased towards near visible
wavelengths, and this characteristics is consistent with the aforementioned BrC absorption spectral
dependence characteristics in Sect 3.1. Actually, the relatively low O/C of Night-OA while high N/C
and H/C ratios (0.04 and 1.89, even higher than those of BBOA which are known composed of complex
nitrogen containing compounds)  was consistent with the features (low O/C, high N/C (ring- and N-
containing structures of NACs) and low hygroscopicity) of aerosols that involved such as nitro-
phenolic compounds (Chen et al., 2022) secondarily formed from reactions of cyclic aromatics that
involve $NO_3$ radical chemistry (Rana and Guzman, 2022;Mayorga et al., 2021) during nighttime which
likely have larger molecular weight  and thereby lower hygroscopicity (Wang et al., 2019a;Price et al.,
2022;Petters et al., 2009). The increased loading of Night-OA would lower organic aerosol
hygroscopicity, which was confirmed previously by Kuang et al. (2021). However, the high N/C of
Night-OA was consistent with its strong absorptivity considering that aerosol absorptions are generally
linked with N-containing components (Qin et al., 2018;Kasthuriarachchi et al., 2020), however, N/C
of 0.04 implies the complex structures of Night-OA components. These results also demonstrate that
NAC are probably key components of the Night-OA factor, considering that biomass burning was
indeed important sources of precursors (Li et al., 2020b;Decker et al., 2019) to form NACs, consistent
with the inference that precursors of Night-OA came from biomass burning. On the basis of limited
existing literatures, Wang et al. (2019c) have summarized the secondary NAC formation pathways of
benzene, toluene, phenol and methylcatechol which represented a significant fraction of biomass
burning VOC emissions (Stockwell et al., 2015), and emphasized that the $NO_3$ radical does not directly
participate in the aqueous phase reactions in NAC formation reaction chains, but first oxidizes these
precursors into intermediate products which then react with $NO_2$ to form nitrophenols and
nitrocatechols that are further oxidized in the particle phase to generate NAC. This gas-phase oxidation
and subsequent particle-phase reaction chains of NAC production may explain the observed Night-OA
formation characteristics that the most-prominent Night-OA formation occurred after the strongest
$NO_3$ radical chemistry when the aerosol liquid water content was abundant, highlighting the great



importance of nighttime multiphase chemistry in Night-OA formation. Aside from the case shown in
Fig.5, much higher correlation coefficient between nighttime average Night-OA mass concentration
and nighttime average $O_x$ concentrations than that of $NO_2$ shown in Fig.S9 for the entire campaign
(0.58 versus 0.39) further stresses that the coordination of night-time gas-phase and aqueous-phase
was responsible for the quick Night-OA formation. In addition, obvious daytime increase in Night-OA
was occasionally observed (Fig.S10 and Fig.S11) as mentioned in Sect.2 when the RH remain high
and $NO_2$ as well as biomass burning emissions increased substantially. This results further emphasizes
that $NO_2$ might play significant roles in aqueous phase reactions of oxidized biomass burning
precursors which might be directly emitted or transformed through active nighttime $NO_3$ chemistry
(Decker et al., 2019).

**4. Environmental and climate significance**
Biomass burning emissions, as the largest sources of primary aerosols and the second largest
sources of VOCs, exert increasingly greater impacts on the regional air quality and global climate
under the context of global warming (Running, 2006). However, the lack of knowledge on the complex
chemical aging of biomass burning plumes during its transport hinders the accurate representation of
biomass burning VOCs and BBOA evolution in air quality and climate models. Kodros et al. (2022)
concluded based on laboratory experiments that dark aging of biomass burning emissions was sensitive
to RH, and their results suggested that the SOA was mainly formed through condensation of gas-phase
oxidation products, however, not ruling out the possibility of heterogeneous oxidations. This study
highlights that nighttime gas-phase chemistry of biomass burning VOC precursors in conjunction with
further aqueous-phase reactions likely contributed substantially to ambient SOA that is highly
absorptive. Although more comprehensive studies about detailed mechanisms are needed and the
robustness of the conclusion in this study needs to be further examined as well as the exact types of
fires during this campaign are not known, still, this finding has important implications for our
understanding on nighttime evolution of biomass burning plumes. Field observations in this study
revealed the abundant existence of highly hygroscopic inorganic components within ambient aerosols
that contributed substantially to aerosol liquid water in ambient air, while aerosols produced in the
chamber of Kodros et al. (2022) were dominated by organics (only with very small amounts of
inorganic nitrate formed during the aging processes) which might have inhibited aqueous reaction



pathways. Inspired by this study, future laboratory studies investigating nighttime aging of biomass
burning plumes should consider not only the high nighttime RH conditions, but also the complex
mixture of background inorganic aerosols with biomass burning emissions when simulating aerosols
within biomass burning plumes.

In addition, biomass burning plumes might form convective clouds under strong ground surface

heating and moisture release (pyrocumulus clouds or pyrocumulonimbus clouds) (Andreae et al.,
2004;Fromm et al., 2006;Cunningham and Reeder, 2009;Lareau and Clements, 2016), or interact with
clouds/fogs during long-range transport (Engelhart et al., 2011). Clouds/fogs provide large amount of
liquid water regardless of day or night, while active photochemistry in biomass burning plumes may
increase $O_3$ and $NO_2$ production (Hecobian et al., 2011;Marufu et al., 2000;Ziemke et al., 2009)
(biomass burning also emits $NO_2$ directly as shown in Fig.45, likely promoting $NO_3$ radical production
(Selimovic et al., 2020) during nighttime. This suggests a potential formation of highly absorptive
SOA in cloud droplets, which results in even browner cloud droplets, impacting cloud optical
properties, cloud lifetime (e.g. promoting cloud burning off effect) and precipitation. Especially, since
violent pyroconvection can penetrate into the stratosphere, darkening of clouds might have regional
impacts on cloud properties as well as radiative balance. Moreover, the biomass burning plumes
transported from remote areas can sometimes mix down into the boundary layer and exert significant
impacts on regional air quality (Wang et al., 2018b). The top of the boundary layer is usually
characterized by high RH conditions due to abundant water vapor but lower temperatures, as well as
high $O_x$ concentrations due to substantial anthropogenic $NO_x$ emissions and active photochemistry,
and thus potentially large amounts of highly absorptive SOA formation could be expected in this kind
of biomass burning plumes.

Overall, results of this study provide important insights into nighttime evolutions of biomass

burning and an uncovered potential secondary BrC formation mechanism that has broad implications
for climate and air quality effects of biomass burning, especially the interaction between biomass
burning plumes with clouds in the aging process during nighttime. However, much more efforts are
still required to further disentangle the complex routes of gas-phase biomass burning VOC oxidation
and subsequent aqueous-phase reactions.




**Data availability**. The data used in this study are available from the corresponding author upon
request Ye Kuang (kuangye@jnu.edu.cn) and Shan Huang (shanhuang_eci@jnu.edu.cn)
**Competing interests**. The authors declare no competing financial interest.

**Author Contributions**. YK conceived this research and wrote the manuscript. SH conducted the
SPAMS measurements and performed the PMF analysis. BL performed the AE33 measurements. MS
and BY planned this campaign.  JL, WH, YP, DC, and DY participated this campaign and provided
full support for the campaign. WX provided insights into reaction mechanism analysis.

**Financial supports.** This work is supported by the National Natural Science Foundation of China
(grant No. 41805109, 41807302), National Key Research and Development Program of China (grant
No. 2017YFC0212803, 2016YFC0202206), Key-Area Research and Development Program of
Guangdong Province (grant No. 2019B110206001), Special Fund Project for Science and Technology
Innovation Strategy of Guangdong Province (grant No.2019B121205004), Guangdong Natural
Science Funds for Distinguished Young Scholar (grant No. 2018B030306037).












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
