# Peer review of "Formation of Highly Absorptive Secondary Brown Carbon Through Nighttime"

_EGUsphere, 2024_

## Author Comment (AC1)

**Responses to anonymous referee #1**

**General comments:**

Ye Kuang et al. present an interesting study on the nighttime formation of secondary organic aerosol (Night-OA) from field observations of biomass burning emissions in the Pearl River Delta region of China. The study highlights the importance of nighttime chemistry in the evolution of biomass burning plumes, particularly the role of NO3 radical chemistry and aqueous-phase processes in the formation of highly light-absorbing organic aerosols. The authors primarily use correlation coefficients to support their hypotheses. However, the study would benefit from additional analysis and discussion. For example, a more detailed investigation into the hypothesized reactions at play through box modeling would be valuable to assess the importance of the specific mechanisms involved in its formation. Within the paper, the authors also spend more time providing detailed explanations on key concepts. Finally, this article would strongly benefit from an English grammar review. I would not recommend this publication until my concerns are addressed. Below are my comments.

**Response:** We sincerely thank the reviewer for acknowledging the value of our study and for providing constructive suggestions. We truly appreciate the time and effort spent reviewing our paper, particularly for those editorial comments.

We fully recognize that exploring the reactions and mechanisms behind the observed nighttime formation of secondary organic aerosols (SOA) from biomass burning precursors would be beneficial. However, due to limitations in our observations, such as the lack of detailed molecular-level measurements of the Night-OA precursors and products, this remains beyond the scope of our current study. The main strength of this work lies in identifying the nighttime formation of secondary brown carbon from multiphase reactions involving biomass burning emissions and preexisting hygroscopic aerosols in the plume. Given the significant gap in field measurements that directly observe nighttime biomass burning plume evolution and confirm the contributions of SOA and brown carbon (BrC) from nighttime aging, providing this key evidence is already a considerable effort. Exploring all aspects of the reaction chains in a single study is not feasible at this stage.

Furthermore, although we have not provided detailed explanations of the reaction chains involved in

Night-OA formation, we believe our findings are still valuable for advancing understanding of the role of nighttime SOA formation in aerosol absorption properties. Additionally, our study paves the way for designing targeted laboratory experiments in the future. For instance, current laboratory studies or box modeling simulations have yet to account for the significant role that preexisting background aerosols play in the aging of diluted biomass burning plumes.

**Major comments:**

**Comment:** This paper introduces a wide variety of scientific hypotheses and reasoning without a proper explanation to the audience (i.e., me).

For example, on page 8, the authors test whether Night-OA was associated with transported plumes containing NOx. They use diurnal variations of Night-OA/CO and Night-OA/NOx ratios to conclude Night-OA is likely formed through secondary processes. Are these separate thoughts? How does the transported NOx get tested and what do the ratios mean with regards to transported NOx?

**Response**: We agree that it was not discussed in a very clear way. We revised this part as:

"Another possibility to consider is whether Night-OA increases could be associated with plumes containing higher $NO_x$ transported from other regions. We investigated the diurnal variations of the Night-OA/CO and Night-OA/$NO_x$ ratios. If Night-OA was transported together with NOx, their ratio would retain characteristics, such as remain near constant. However, persistent increases in Night-OA/CO and Night-OA/$NO_x$ ratios were observed when significant Night-OA formation began. This suggests that Night-OA is likely formed through secondary processes that would result in ratio increase of Night-OA to both $NO_x$ and CO, consistent with that it was also correlated with nitrate (R=0.67)."

**Comment**: Another example is provided in the minor comments section, line 176. It would benefit both the reader and the author to expand on these ideas and explain them.

**Response**: This part about aBBOA was carefully discussed in the revised manuscript.

"Given that PMF analysis is fundamental to our study, the mass spectral profiles of factors are provided in Fig. S1, and key aspects of the resolved results are explained here, particularly concerning

naming of aBBOA. In previous studies (Kuang et al., 2021;Luo et al., 2022), we already realized that the correlation between aBBOA and $C_6H_2NO_4^+$ was actually weak (R=0.31), suggesting that it might not fully be constituted of aging products of primary BBOA considering its O/C was even lower than that of BBOA. The following clues demonstrate that aBBOA is mostly secondary and formed from the gas-phase reactions: (1) aBBOA exhibited similar diurnal behavior to LOOA, showing clear daytime photochemical production (Fig.S1); (2) The increase in aBBOA loading enhances organic aerosol hygroscopicity despite its low O/C ratio, as demonstrated by Kuang et al. (2021). In contrast, primary organic aerosols have not been observed to enhance overall organic aerosol hygroscopicity (Kuang et al., 2020b;Kuang et al., 2024;Tao et al., 2024); (3) aBBOA primarily added mass to the condensation mode diameter range (see discussions on OA factor size distributions in the supplement of Luo et al. (2022)), suggesting that aBBOA forms from vapor condensation after gas-phase reactions (Kuang et al., 2020a). Furthermore, the following facts indicate that aBBOA is formed from gas-phase reactions of biomass burning precursors: (1) An evening peak in aBBOA around 19:00 (local time) occurs just after the peak of BBOA emissions, as shown in Fig. S1, although a small portion of aBBOA may be directly emitted (the increase in aBBOA during BBOA emissions accounts for an average of 8% of the mass increase in biomass burning events, Fig. 3 of Luo et al. (2022)); (2) The daytime formation of aBBOA is correlated with the biomass burning intensity from the previous night (Wu et al., 2024); (3) The high absorptivity of aBBOA (introduced in Sect 3.1) which is comparable to SOA formed from biomass burning precursors (Saleh et al., 2013). As demonstrated in previous indoor experiments on biomass burning emissions (Yee et al., 2013; Ahern et al., 2019), SOA formation from gas-phase reactions could have a low O/C during short oxidation periods, which help explain the low O/C of aBBOA. Given these convincing, the original designation of aBBOA may not be appropriate. Therefore, we have renamed this factor as BB-SOA to reflect that it is an SOA factor related to biomass burning emissions."

**Comment**: Another example is explaining to the reader how the carbon signals from SP-AMS were used to retrieve the shape of BC mass size distribution. The supplement of Luo et al. (2022) shows a simple formula (equation 8). Why not include it for the readers?

**Response**: Added.

"The real-time measured carbon fragments (C$_x$) distributions by the SP-AMS were therefore used to distribute the total BC mass to different diameter bins (Eq.8 of the supplement in Luo et al. (2022)) to calculate RAAE ($\lambda$) as introduced in Luo et al. (2022)."

**Comment**: What is the multivariate linear regression method for reporting mass absorption efficiencies? Please define mass absorption efficiency.

**Response:** Revised as the following:

"The average mass absorption efficiencies (MAEs, defined as absorption coefficient per unit mass, m$^2$/g) of different OA factors are retrieved using multivariate linear regression method which were commonly used for this purpose (de Sá et al., 2019;Kasthuriarachchi et al., 2020) though bear uncertainties. The multivariate linear regression method was expressed as $\sigma_{BrC}$ ($\lambda$)= [HOA]$\times$ MAE$_{HOA}(\lambda)$ + [MOOA]$\times$MSE$_{MOOA}(\lambda)$+[BBOA]$\times$MSE$_{BBOA}(\lambda)$ + [Night-OA]$\times$MSE$_{Night-OA}(\lambda)$ + [BB-SOA]$\times$MSE$_{BB-SOA}(\lambda)$ where [OA factor] represents mass concentration and $\lambda$ represents optical wavelength. "

**Comment:** The authors spend much of their time in the second half of the paper discussing implications and suggestions as to how all these compounds interact with one another. It is not very convincing to me. I would like to see the field measurements, like those in Figure 5, put into a box model to see if these proposed reactions and mechanisms can reproduce these mixing ratios. Additionally, they can quantitatively compare the reaction rates of these different pathways.

**Response:** Thanks for your suggestion. We really understand that it would be beneficial to explore reactions and mechanisms behand the observed nighttime formation of SOA from biomass burning precursors. It is beyond our capability with limited observations, for example, lack of detained precursor and product measurements of the Night-OA factor in a molecular level. The real strength of this work is the finding of nighttime secondary brown carbon formation from multiphase reactions of biomass burning emissions, considering the stage that field measurements that observed nighttime evolutions of biomass burning plumes and directly confirmed significant contributions of SOA and

BrC from nighttime aging of biomass burning emissions are highly lacking. In addition, we do think discussions about how Night-OA formation related with different factors would be helpful for both understanding the importance of night SOA formation from biomass burning emissions in aerosol absorptions and paving the way for designing targeted laboratory experiments in future. For example, current laboratory studies or box model simulations have not considered the important role of preexisting background aerosols played in the aging of biomass burning plumes. Preexisting background aerosol are generally much hygroscopic that aerosols directly emitted from biomass burning emissions. Clues and arguments like this are indeed important for future laboratory studies that aim to disentangle important reactions and mechanisms that matter for biomass burning aging and corresponding SOA formation.

**Technical points:**

**Comment:** Line 73: I would adjust this sentence to say, "…Palm et al. (2020) observed that daytime oxidation of emitted phenolic compounds contributed a majority to BBSOA formation from direct gas emissions, with products…" In Figure 4 of Palm et al. (2020), he demonstrates that the majority of BBSOA (87%) is from evaporation of BBPOA.

**Response**: Revised accordingly.

**Comment:** In section 2.1, could you please describe what specific instruments were used to measure RH, temperature, and wind speed and direction ?.

**Response**: Added, as the following:

"such as RH, temperature, wind speeds and directions (Model WXT520, Vaisala, Finland)"

**Comment:** Line 152: What are the chemicals composing the night-OA factor? The BBOA factor? Did other variables correlate in time with the night-OA factor? What other chemicals make up the aged BBOA factor? It sounds like it is a misnomer if nitrocatechol is the only "aged" compound. I don't even think it is aged by that much. It is only 2 reactions away from the primary biomass burning

compound catechol. Granted, you do discuss this misnomer in line 163 but continue to use this label throughout the paper. Perhaps oBBOA (oxidized BBOA) would be better. I also understand that at line 212, you decide to keep the labels for consistency with previous work. I would argue that the names should change since you know they are not completely appropriate.

**Response:** We have more carefully discussed Night-OA and aBBOA factor in the revised manuscript in Sect 2.1 and Sect 3.1.

For aBBOA:

"Given that PMF analysis is fundamental to our study, the mass spectral profiles of factors are provided in Fig. S1, and key aspects of the resolved results are explained here, particularly concerning naming of aBBOA. In previous studies (Kuang et al., 2021;Luo et al., 2022), we already realized that the correlation between aBBOA and $C_6H_2NO_4^+$ was actually weak (R=0.31), suggesting that it might not fully be constituted of aging products of primary BBOA considering its O/C was even lower than that of BBOA. The following clues demonstrate that aBBOA is mostly secondary and formed from the gas-phase reactions: (1) aBBOA exhibited similar diurnal behavior to LOOA, showing clear daytime photochemical production (Fig.S1); (2) The increase in aBBOA loading enhances organic aerosol hygroscopicity despite its low O/C ratio, as demonstrated by Kuang et al. (2021). In contrast, primary organic aerosols have not been observed to enhance overall organic aerosol hygroscopicity (Kuang et al., 2020b;Kuang et al., 2024;Tao et al., 2024); (3) aBBOA primarily added mass to the condensation mode diameter range (see discussions on OA factor size distributions in the supplement of Luo et al. (2022)), suggesting that aBBOA forms from vapor condensation after gas-phase reactions (Kuang et al., 2020a). Furthermore, the following facts indicate that aBBOA is formed from gas-phase reactions of biomass burning precursors: (1) An evening peak in aBBOA around 19:00 (local time) occurs just after the peak of BBOA emissions, as shown in Fig. S1, although a small portion of aBBOA may be directly emitted (the increase in aBBOA during BBOA emissions accounts for an average of 8% of the mass increase during biomass burning events, Fig. 3 of Luo et al. (2022)); (2) The daytime formation of aBBOA is correlated with the biomass burning intensity from the previous night (Wu et al., 2024); (3) The high absorptivity of aBBOA (introduced in Sect 3.1) which is comparable to SOA formed from biomass burning precursors (Saleh et al., 2013). As demonstrated in previous indoor experiments on

biomass burning emissions (Yee et al., 2013; Ahern et al., 2019), SOA formation from gas-phase reactions could have a low O/C during short oxidation periods, which help explain the low O/C of aBBOA. Given these convincing, the original designation of aBBOA may not be appropriate. Therefore, we have renamed this factor as BB-SOA to reflect that it is an SOA factor related to biomass burning emissions."

For Night-OA:

"The Night-OA was named because of its prominent night increase (Fig.3a) and previously speculated as secondary because of its tight correlation with nitrate. However, Night-OA has a relatively low O/C ratio of 0.32, raising the question of whether it originates from primary emissions or secondary formation. As discussed in Luo et al. (2022), traffic, cooking (The HOA and cooking-related OA and were not separated in the PMF results although the hydrocarbon-like factor was named HOA as discussed in Kuang et al. (2021)), and biomass burning are likely the dominant primary sources during this campaign. If Night-OA were a primary source, it would be expected to increase alongside other primary sources. We identified most Night-OA increase events and examined their correlation with variations in other primary sources, as shown in Fig. 3b-e. This analysis reveals that Night-OA increases were typically observed after sunset, though occasionally during the daytime. Night-OA increases showed weak correlations with changes of CO (R=0.25), HOA (R=0.1), and BBOA (R=-0.15), but a moderate correlation with $NO_x$ (R=0.55). During significant biomass burning events (indicated by substantial BBOA increases), the concentration of Night-OA decreased on average (Fig. 3 of Luo et al. (2022)), suggesting that Night-OA is unlikely to be emitted from biomass burning. We also identified all significant HOA increase events that did not coincide with biomass burning and analyzed the average HOA increase and variations in other aerosol components (Fig. S2). It shows that, despite significant HOA increases, the average mass concentration of Night-OA remained almost unchanged, indicating that Night-OA is also unlikely to originate from HOA-associated emissions. Therefore, the weak but positive correlations between Night-OA and HOA as well as CO are likely associated with the accumulation characteristics of primary emissions after sunset. The higher correlations between Night-OA and $NO_x$ may also result from the accumulation of $NO_x$ starting in the afternoon when photochemical depletion is weaker. Another possibility to consider is whether Night-

OA increases could be associated with plumes containing higher $NO_x$ transported from other regions. We investigated the diurnal variations of the Night-OA/CO and Night-OA/$NO_x$ ratios. If Night-OA was transported together with NOx, their ratio would hold characteristics such as remain near constant. However, persistent increases in Night-OA/CO and Night-OA/$NO_x$ ratios were observed when significant Night-OA formation began. This suggests that Night-OA is likely formed through secondary processes that would result in ratio increase of Night-OA to both $NO_x$ and CO, consistent with that it was also correlated with nitrate (R=0.67). The low O/C of Night-OA, still higher than that of the primary factor HOA, was determined by a high amount of $C_xH_y^+$ ions. However, the Night-OA was also characterized with significant intensity of oxidation tracers $C_2H_3O^+$ and $CO_2^+$, suggesting that Night-OA was oxidation products with low oxidation state during the nighttime. A similar situation was previously found in a study at Bakersfield of USA in which Liu et al. (2012) identified SOA factors as alkane-SOA and aromatic-SOA with moderate O/C (0.27-0.36). Meanwhile, $NO_x$ potentially promoted its formation, given the highest N/C ratio of Night-OA among all resolved factors. This will be discussed further in Sect 3.2."

We agree. We rename this factor as BB-SOA, and the following sentence is added in the paragraph that discusses the name of aBBOA:

"Given all these clues, the naming of aBBOA might not be appropriate, we rename this factor as BB-SOA in the following text considering it is a SOA factor that relates with biomass burning emissions."

**Comment:** Line 168: How does it make sense that the evening peak of aBBOA correlating with the aBBOA noon peak in the next day means it could be emitted from biomass burning? Or did you mean the BBOA noon peak? The wording is unclear.

**Response:** Many thanks for pointing this out. This is confusing. We want to tell that aBBOA formation in next day is related with night BBOA emissions, this part of discussions is revised as the following:

"(2) The daytime aBBOA formation is correlated with the biomass burning intensity during the night before (Wu et al., 2024);"

**Comment:** Line 176: Please further explain why the hypothesis that aBBOA most likely originated

from the gas-phase oxidation of biomass burning emitted VOC precursors is supported by aBBOA loading enhancing hygroscopicity. The connection is not clear in the text.

**Response**: The following two facts demonstrate that aBBOA (now called BB-SOA) is mostly secondary formed from gas-phase reactions: (1) aBBOA exhibited similar diurnal behavior to LOOA, showing clear daytime photochemical production; (2) The fact that increases in aBBOA loading enhance organic aerosol hygroscopicity despite its low O/C, as demonstrated by Kuang et al. (2021), whereas primary organic aerosols have not been observed to enhance overall organic aerosol hygroscopicity (Kuang et al., 2020b;Kuang et al., 2024;Tao et al., 2024);

While the following three facts support that aBBOA is related with biomass burning emissions: (1) A aBBOA peak appeared just after the peak of the biomass burning emissions; (2) The daytime aBBOA formation is correlated with the biomass burning intensity during the night before (Wu et al., 2024); (3) The high absorptivity of aBBOA which is close to SOA formed from biomass burning precursors (Saleh et al., 2013).

To make these points clearer, the paragraph was revised as the following:

"Given that PMF analysis is fundamental to our study, the mass spectral profiles of factors are provided in Fig. S1, and key aspects of the resolved results are explained here, particularly concerning naming of aBBOA. In previous studies (Kuang et al., 2021;Luo et al., 2022), we already realized that the correlation between aBBOA and $C_6H_2NO_4^+$ was actually weak (R=0.31), suggesting that it might not fully be constituted of aging products of primary BBOA considering its O/C was even lower than that of BBOA. The following clues demonstrate that aBBOA is mostly secondary and formed from the gas-phase reactions: (1) aBBOA exhibited similar diurnal behavior to LOOA, showing clear daytime photochemical production (Fig.S1); (2) The increase in aBBOA loading enhances organic aerosol hygroscopicity despite its low O/C ratio, as demonstrated by Kuang et al. (2021). In contrast, primary organic aerosols have not been observed to enhance overall organic aerosol hygroscopicity (Kuang et al., 2020b;Kuang et al., 2024;Tao et al., 2024); (3) aBBOA primarily added mass to the condensation mode diameter range (see discussions on OA factor size distributions in the supplement of Luo et al. (2022)), suggesting that aBBOA forms from vapor condensation after gas-phase reactions (Kuang et al., 2020a). Furthermore, the following facts indicate that aBBOA is formed from gas-phase reactions

of biomass burning precursors: (1) An evening peak in aBBOA around 19:00 (local time), occurring just after the peak of BBOA emissions, as shown in Fig. S1, although a small portion of aBBOA may be directly emitted (the increase in aBBOA during BBOA emissions accounts for an average of 8% of the mass increase during biomass burning events, Fig. 3 of Luo et al. (2022)); (2) The daytime formation of aBBOA is correlated with the biomass burning intensity from the previous night (Wu et al., 2024); (3) The high absorptivity of aBBOA (introduced in Sect 3.1) which is comparable to SOA formed from biomass burning precursors (Saleh et al., 2013). As demonstrated in previous indoor experiments on biomass burning emissions (Yee et al., 2013; Ahern et al., 2019), SOA formation from gas-phase reactions could have a low O/C during short oxidation periods, which help explain the low O/C of aBBOA. Given these convincing, the original designation of aBBOA may not be appropriate. Therefore, we have renamed this factor as BB-SOA to reflect that it is an SOA factor related to biomass burning emissions. ”

**Comment:** Figure 1: For the color of the scatter plots, is it the average time of the period of observation (e.g., 12 AM), or is it the average length of time of night-OA increase cases? The text implies the former, but it is not clear in the units (e.g., Local Time).

**Response:** It is the average time of the period of observation, "local time" is added. In the revised manuscript, we adjusted the order of figures according to reviewers' comments, so this figure is shown as Figure 3 in the latest version.

[Figure]

**Figure 3. (a)** Diurnal variations of nitrate, Night-OA and BBOA; **(b-e)** Relations between increases of Night-OA and increases of CO, BBOA, HOA and $NO_x$ for identified Night-OA increase cases; colors of scatter plots represent the average local time of Night-OA increase cases; **(f)** Diurnal variations of ratios Night-OA/CO, Night-OA/$NO_x$ in the left axis and $NO_x$ in the right axis.

**Comment:** Line 182: Please define COA.

**Response:** Revised as "The HOA and cooking-related OA"

**Comment:** Line 205: Are the CxHy+ ions anticipated to be fragments of oxidized materials? Why?

**Response:** Yes, we think $C_xH_y^+$ ions are likely anticipated to be fragments of oxidized products. As we explained in previous responses, Night-OA is likely formed through secondary processes, while significant intensity of oxidation tracers $C_2H_3O^+$ and $CO_2^+$ and a high amount of $C_xH_y^+$ ions suggests that Night-OA was oxidation products with low oxidation state during the nighttime. The abundant $C_xH_y^+$ ions explained the low O/C (0.32) of Night-OA, which is consistent with the findings at

Bakersfield of USA in which Liu et al. (2012) identified SOA factors as alkane-SOA and aromatic-SOA with moderate O/C (0.27-0.36). Considering abundant alkyl- and aryl-contained species emitted from biomass burning, C-H bonds and aromatic rings could still exist in low-oxidation-state products. Their fragments could appear as $C_xH_y^+$ in Night-OA mass spectrum.

**Comment:** Line 228: Why is the formula's exponent AAEBC,950-880 x RAAE(λ), when the denominator of RAAE is already AAEBC,950-880? Shouldn't it be simplified to AAEBC,λ-880?

**Response:** Here, $AAE_{BC,\lambda\text{-}880}$ equals to $AAE_{BC,950\text{-}880}$ x $R_{AAE}(\lambda)$. We use this expression because of the need to introduce $R_{AAE}(\lambda)$ values.

**Comment:** Line 252: Please include the correlation coefficient figures in, at the very least, the supplement. However, wouldn't we expect the relationship between BBOA and σBrC to decrease with wavelength increases? BrC is not very absorptive at larger wavelengths.

**Response:** Thank you for this comment. The figure was added in the supplement as Figure S2 and shown below.

The decrease of correlation coefficients was explained by the observed secondary BrC formation. We didn't see why BrC absorbs less would result in higher correlation with BBOA. The correlation is more influenced by BrC sources, not absorptive capability.

[Figure]

**Figure S2. (a)** Box-and-whisker plots of BrC absorption fractions at different wavelengths; **(b-f)** Correlations between BrC absorptions at 370 nm, 470 nm, 520 nm, 590 nm and 660 nm and BBOA.

**Comment:** Line 254: Could you show us a plot of BBOA vs σBrC,370 colored in two colors representing daytime and nighttime?

**Response:** Thanks for the comment. The plot of BBOA vs $\sigma_{BrC,370}$ in daytime and nighttime is shown below. Actually, Figure 1a has presented the detailed co-variation of BBOA and $\sigma_{BrC,370}$ during daytime

and nighttime (grey shadows) in the whole campaign. Also, the diurnal variations of σBrC,370 vs BBOA cloud be clearly indicated by the average σBrC,370/BBOA ratio shown in Fig.2a. This sentence is revised as the following:

"In addition, as shown in Fig.1a, coordinal variations between BBOA and $\sigma_{BrC,370}$ are usually seen during daytime especially during the dusk BBOA spike periods, however, the $\sigma_{BrC,370}$ sometimes deviates substantially from BBOA variations during the nighttime (gray areas in Fig.1). The average diurnal variations of both $\sigma_{BrC,370}$ and the ratio $\sigma_{BrC,370}$/BBOA is presented in Fig.2a, and quick $\sigma_{BrC,370}$/BBOA increases were observed during nighttime before 06:00 LT."

**Comment:** Line 259: "and differ much at different wavelengths." I do not think the figure demonstrates anything about different wavelengths. Please remove this phrase.

**Response:** Sorry for the confusing statement. The analysis was provided by newly-added Fig.S2 as suggested in the previous comment. To make it clearer, we revised this sentence as:

"These results demonstrate that organic aerosol components other than BBOA also contribute substantially to BrC absorption and differ much at different wavelengths as indicated by distinct correlations between BBOA and $\sigma_{BrC,370}$ at different wavelengths."

**Comment:** Line 290: How does the multivariate linear regression method translate to different OA factors, such as BBOA and Night-OA? I think this needs more of an explanation.

**Response:** To make this clearer, this part was revised as the following:

"The aforementioned multivariate linear regression method is thus also used for retrieving MAEs of BBOA and Night-OA at wavelengths of 470 nm, 520 nm, 590 nm and 660 nm. The overall fitting performance of using retrieved MAE values at multiple wavelengths are shown in Fig.S3."

The method was introduced in paragraph 2 of Sect 3.1:

"The average mass absorption efficiencies (MAEs, defined as absorption coefficient per unit mass, m2/g) of different OA factors are retrieved using multivariate linear regression method which were commonly used for this purpose (de Sá et al., 2019;Kasthuriarachchi et al., 2020) though bear uncertainties. The multivariate linear regression method was expressed as $\sigma_{BrC}(\lambda)=$ [HOA] $\times$

$MAE_{HOA}(\lambda) + [MOOA] \times MSE_{MOOA}(\lambda) + [BBOA] \times MSE_{BBOA}(\lambda) + [Night\text{-}OA] \times MSE_{Night\text{-}OA}(\lambda) + [BB\text{-}SOA] \times MSE_{BB\text{-}SOA}(\lambda)$ where [OA factor] represents mass concentration and $\lambda$ represents optical wavelength."

**Comment:** Line 304: Can you explain why you think Night-OA is evaporated during the daytime?

**Response:** Thanks for pointing this out. We do not think that Night-OA is certainly evaporated during the daytime, this is why we stated like this in the discussion "The phenomenon that Night-OA daytime loss is more correlated with RH decrease implied that Night-OA possibly co-evaporated with water vapor as RH decrease. This might serve weakly but still another supporting clue for that Night-OA were likely formed through aqueous pathways and maybe reversible." To avoid misleading, we have revised this sentence as the following:

"Night-OA factor is characterized by its rapid nighttime formation and quick daytime loss (through such as repartitioning, photodegradation, etc.)."

**Comment:** Line 319: What do you mean that "to be Lagrangian, air after midnight might differ with those before midnight?" This is a confusing sentence and I don't understand why midnight is such a special time for air. Later on the dividing time for Figure 4a is 10 PM.

**Response:** we mention "Lagrangian" to highlight that air is not moving continuously, is not stagnant, while the observation location is fixed, and we use example comparison between after midnight and before midnight to stress on the air difference. To make this clearer, we revised this part as the following:

"Indeed, the air is always in motion and behaves in a Lagrangian manner. Although the observation location is fixed, the information collected at different times corresponds to signals from different air parcels. However, if the biomass burning events are regionally representative, the relationship between the observed increase in Night-OA signals after midnight and the biomass burning emission signals before midnight may still provide valuable insights."

**Comment:** Line 351: There is no section 1.2 in your supplement.

**Response:** Sorry for this typo. We have corrected it to be section 1.1.

**Comment:** I don't think Figure S6 shows that the HOA daytime loss is due to evaporation as it only shows that it's not dilution.

**Response:** Agree. We revised this as the following:

"In addition, as shown in Fig.S6, the Night-OA decreased quickly during daytime, which is beyond the dilution effect of boundary layer development (indicated by rapid decrease of Night-OA/CO as shown in Fig.S6). The substantial daytime loss of Night-OA which might be caused by several processes, such as partitioning evaporation, photodegradation (Wang et al., 2023) and chemical transformations, etc."

**Comment:** Line 452: Biomass burning emissions are not the largest sources of primary aerosols. Sea salt emissions are.

**Response:** Thanks for pointing this out. It was revised as:

"as one of the largest sources"

**Editorial comments:**

**Comment:** Throughout the manuscript: I would take some time to look at sentences that are more than three lines long. These sentences either need to be simplified or broken up into multiple sentences. I found it distracting.

Throughout the manuscript: There are a significant amount of grammar mistakes that need to be addressed. I would suggest having an editor read through the manuscript just for grammar improvements.

**Response:** We truly appreciate your valuable comments in both scientific and editorial aspects. We carefully addressed your comments and improved the grammar and text. In addition, we edited the manuscript with the assistance of ChatGPT to ensure clarity and precision in presenting our ideas.

**Comment:** Throughout the manuscript: NOx should be $NO_x$.
**Response:** Revised.

**Comment:** Lines 32 and 33: "Our results demonstrate that the formation of Night-OA appeared high

dependence on both…" should perhaps be "Our results demonstrate that the formation of Night-OA appeared to have high dependence on both…".

**Response:** Revised.

**Comment:** Line 44: "…water abundant pyroconvection cloud" should perhaps be "…water-abundant pyroconvective clouds."

**Response:** Revised.

**Comment:** Line 61: Fires also emit methane, so you can just say "primary organic aerosols and volatile organic compounds.".

**Response:** Revised.

**Comment:** Line 72: "which found that the formed SOA" should be "and those investigations found that the formed SOA…".

**Response:** Revised.

**Comment:** Line 84: "…using NO3 as oxidant…" should be "…using NO3 as the oxidant…" and "…biomass burning emissions related SOA formations…" should be "… SOA formation from biomass burning…".

**Response:** Revised.

**Comment:** Line 92: "Nevertheless, field measurements that observed nighttime evolutions of biomass burning plumes… are highly in lack," should be "Nevertheless, field measurements that observe nighttime evolutions of biomass burning plumes… are highly lacking.".

**Response:** Revised.

**Comment:** Please break the following into two sentences. "…contributing greatly to light absorption (Lin et al., 2017;Bluvshtein et al., 2017). Based on this study, it was hypothesized…".

**Response:** Revised.

**Comment:** Line 101: "…most of previous laboratory…" should be "…most previous laboratory…".

**Response:** Revised.

**Comment:** Line 104: "…could not conclude what roles RH was…" should be "…could not conclude what role RH played.".

**Response:** Revised.

**Comment:** Line 107: "Therefore, how nighttime NO3 radical chemistry coordinates with aerosol

aqueous or heterogenous reactions under high nighttime RH conditions to affect SOA and BrC formations remains unexplored, which is a substantial knowledge gap in the research field of nighttime chemical transformation of biomass burning emissions and its role in SOA and secondary BrC formations," should be "Therefore, how nighttime NO3 radical chemistry coordinates with heterogenous reactions under high RH conditions to affect SOA and BrC formations remains unexplored. Compounding this knowledge gap is how biomass burning emissions contribute to SOA and secondary BrC formation."

**Response:** Revised.

**Comment:** Line 125: "…Peral River Delta…" to "Pearl River Delta.".

**Response:** Revised.

**Comment:** Line 129: Relative humidity was already defined as RH earlier.

**Response:** Revised.

**Comment:** Line 139: "More details on the site and set-up of instruments please…" should be "…For more details on the site and set-up of instruments, please…"

**Response:** Revised

**Comment:** Line 143: "Source identification of organic aerosols was performed using the commonly used positive matrix factorization (PMF), two primary OA factors and four secondary OA factors are identified, and the determination of PMF factors are thoroughly discussed in Luo et al. (2022)." should be "Source identification of organic aerosols was performed using the commonly used positive matrix factorization (PMF). Two primary OA factors and four secondary OA factors were identified and determined from PMF as thoroughly discussed in Luo et al. (2022)."

**Response:** Revised.

**Comment:** Line 149: "The four SOA factors including more oxygenated…" should be "The four SOA factors include more oxygenated…"

**Response:** Revised.

**Comment:** Line 158: "until 1th of November" should be "until the 1st of November." Also, "until 18th of November" should be "until the 18th of November."

**Response:** Revised.

**Comment:** Line 207: "Similar situation was previously found in study…" should be "A similar situation was previously found in a study…"

**Response:** Revised

**Comment:** Line 211: "In summary, both aBBOA and Night-OA are not likely primary, while the naming of…" should be "In summary, both aBBOA and Night-OA are not likely primary. While the naming of…"

**Response:** This sentence is revised as "Given this, the naming of aBBOA might not be appropriate, we rename this factor as BB-SOA considering it is a SOA factor that relates with biomass burning.".

The paragraph here about Night-OA in the previous manuscript is moved to Sect 3.1 according to suggestion of reviewer#2.

**Comment:** Line 216: "Therefore, the details about discussions of this method please refer to Luo et al. (2002), and we only introduce…" should be "The details of this method can be found in Luo et al. (2022). We only introduce…"

**Response:** Revised.

**Comment:** Line 229: "As the sophisticated discussions presented in Luo et al. (2022), variations of many factors… might influences the magnitudes…" should perhaps be "The sophisticated discussions presented in Luo et al. (2022) consider that variations… might influence the magnitudes…"

**Response:** Revised as "As the sophisticated discussions made in Luo et al. (2022), variations of many factors…".

**Comment:** Line 249: "Timeseries of retrieved σBrC,370 is shown…" should be "Timeseries of retrieved σBrC,370and BBOA are shown…"

**Response:** Revised.

**Comment:** Line 254: "BBOA σBrC,370" should be "BBOA and σBrC,370"

**Response:** Revised.

**Comment:** Line 257: "Increase" should be "increases"

**Response:** Revised.

**Comment:** Line 269: "…low absorptivity of LOOA, thus MAE of LOOA is treated as zero." should be "…low absorptivity of LOOA. Thus MAE of LOOA is treated as zero."

**Response:** Revised.

**Comment:** Line 282: "nm are shown in Fig.S4, it tells that Night-OA…" should be "nm are shown in Fig. S4, demonstrating that Night-OA…"

**Response:** Revised.

**Comment:** Line 295: I can't tell which order these four values are presented. Is it BBOA $AAE_{370-470}$, BBOA $AAE_{470-590}$, Night-OA $AAE_{370-470}$, Night-OA $AAE_{470-590}$? Or is it BBOA $AAE_{370-470}$, Night-OA $AAE_{370-470}$, BBOA $AAE_{470-590}$, Night-OA $AAE_{470-590}$?

**Response:** We revised this as:

"retrieved $AAE_{370-470}$, $AAE_{470-590}$ for BBOA are 3 and 5.9, for Night-OA are 1.3 and 6".

**Comment:** Line 302: "increased during the nighttime, while usually decreased…" should be "increased during the nighttime and usually decreased…"

**Response:** Revised.

**Comment:** Line 323: "correlated with BBOA before the night…" should be "correlated with BBOA from the night before…"

**Response:** Revised.

**Comment:** Figure 4a: What are the units on the OX,LT20-06? Please add ppb to the figure, not just the figure description.

**Response:** Added.

**Comment:** Line 346: "diameter range of > 300 nm occurred even all other SOA factors…" should be "diameter range of > 300 nm occurred. Even all other SOA factors…"

**Response:** Revised.

**Comment:** Figure S7: What are the units of NO2 + O3? However, I don't think you talked about NO2 + O3 in the main manuscript. Perhaps the color should be removed.

**Response:** Colors removed.

**Comment:** Line 379: "radical do not photolyze, NO reacts rapidly…" should be "radical do not photolyze. NO reacts rapidly…"

**Response:** Revised.

**Comment:** Line 386: "…this dilution effects…" should be "…this dilution effect…"

**Response:** Revised.

**References:**

de Sá, S. S., Rizzo, L. V., Palm, B. B., Campuzano-Jost, P., Day, D. A., Yee, L. D., Wernis, R., Isaacman-VanWertz, G., Brito, J., Carbone, S., Liu, Y. J., Sedlacek, A., Springston, S., Goldstein, A. H., Barbosa, H. M. J., Alexander, M. L., Artaxo, P., Jimenez, J. L., and Martin, S. T.: Contributions of biomass-burning, urban, and biogenic emissions to the concentrations and light-absorbing properties of particulate matter in central Amazonia during the dry season, Atmos. Chem. Phys., 19, 7973-8001, 10.5194/acp-19-7973-2019, 2019.

Kasthuriarachchi, N. Y., Rivellini, L.-H., Adam, M. G., and Lee, A. K. Y.: Light Absorbing Properties of Primary and Secondary Brown Carbon in a Tropical Urban Environment, Environmental science & technology, 54, 10808-10819, 10.1021/acs.est.0c02414, 2020.

Kuang, Y., He, Y., Xu, W., Yuan, B., Zhang, G., Ma, Z., Wu, C., Wang, C., Wang, S., Zhang, S., Tao, J., Ma, N., Su, H., Cheng, Y., Shao, M., and Sun, Y.: Photochemical Aqueous-Phase Reactions Induce Rapid Daytime Formation of Oxygenated Organic Aerosol on the North China Plain, Environmental science & technology, 54, 3849-3860, 10.1021/acs.est.9b06836, 2020a.

Kuang, Y., Xu, W., Tao, J., Ma, N., Zhao, C., and Shao, M.: A Review on Laboratory Studies and Field Measurements of Atmospheric Organic Aerosol Hygroscopicity and Its Parameterization Based on Oxidation Levels, Current Pollution Reports, 10.1007/s40726-020-00164-2, 2020b.

Kuang, Y., Huang, S., Xue, B., Luo, B., Song, Q., Chen, W., Hu, W., Li, W., Zhao, P., Cai, M., Peng, Y., Qi, J., Li, T., Wang, S., Chen, D., Yue, D., Yuan, B., and Shao, M.: Contrasting effects of secondary organic aerosol formations on organic aerosol hygroscopicity, Atmos. Chem. Phys., 21, 10375-10391, 10.5194/acp-21-10375-2021, 2021.

Kuang, Y., Xu, W., Tao, J., Luo, B., Liu, L., Xu, H., Xu, W., Xue, B., Zhai, M., Liu, P., and Sun, Y.: Divergent Impacts of Biomass Burning and Fossil Fuel Combustion Aerosols on Fog-Cloud Microphysics and Chemistry: Novel Insights From Advanced Aerosol-Fog Sampling, Geophysical Research Letters, 51, e2023GL107147, https://doi.org/10.1029/2023GL107147, 2024.

Liu, S., Ahlm, L., Day, D. A., Russell, L. M., Zhao, Y., Gentner, D. R., Weber, R. J., Goldstein, A. H., Jaoui, M., Offenberg, J. H., Kleindienst, T. E., Rubitschun, C., Surratt, J. D., Sheesley, R. J., and Scheller, S.: Secondary organic aerosol formation from fossil fuel sources contribute majority of summertime organic mass at Bakersfield, J. Geophys. Res. - Atmos., 117, https://doi.org/10.1029/2012JD018170, 2012.

Luo, B., Kuang, Y., Huang, S., Song, Q., Hu, W., Li, W., Peng, Y., Chen, D., Yue, D., Yuan, B., and Shao, M.: Parameterizations of size distribution and refractive index of biomass burning organic aerosol with black carbon content, Atmos. Chem. Phys., 22, 12401-12415, 10.5194/acp-22-12401-2022, 2022.

Saleh, R., Hennigan, C. J., McMeeking, G. R., Chuang, W. K., Robinson, E. S., Coe, H., Donahue, N. M., and Robinson, A. L.: Absorptivity of brown carbon in fresh and photo-chemically aged biomass-burning emissions, Atmos. Chem. Phys., 13, 7683-7693, 10.5194/acp-13-7683-2013, 2013.

Tao, J., Luo, B., Xu, W., Zhao, G., Xu, H., Xue, B., Zhai, M., Xu, W., Zhao, H., Ren, S., Zhou, G., Liu, L., Kuang, Y., and Sun, Y.: Markedly different impacts of primary emissions and secondary aerosol formation on aerosol mixing states revealed by simultaneous measurements of CCNC, H(/V)TDMA, and SP2, Atmos. Chem. Phys., 24, 9131-9154, 10.5194/acp-24-9131-2024, 2024.

Wang, Y., Qiu, T., Zhang, C., Hao, T., Mabato, B. R. G., Zhang, R., Gen, M., Chan, M. N., Huang, D.

D., Ge, X., Wang, J., Du, L., Huang, R.-J., Chen, Q., Hoi, K. I., Mok, K. M., Chan, C. K., and Li, Y. J.: Co-photolysis of mixed chromophores affects atmospheric lifetimes of brown carbon, Environmental Science: Atmospheres, 3, 1145-1158, 10.1039/D3EA00073G, 2023.

Wu, L., Huang, S., Liu, Y., Song, Q., Hu, W., Chen, W., Kuang, Y., Wang, X., Li, W., Peng, Y., Chen, D., Yue, D., Song, W., Yuan, B., Wang, X., and Shao, M.: Source, formation mechanism and inhalation deposition flux of organic aerosols in urban and rural areas of the Pearl River Delta, Acta Scientiae Circumstantiae, 44, 15-28, 10.13671/j.hjkxxb.2023.0194, 2024.

---

## Author Comment (AC2)

**Responses to anonymous referee #2**

**General comments:**

**Comment:**   In this manuscript, the authors present evidence from field measurements that the nighttime formation of light-absorbing secondary organic aerosol from biomass burning emissions may involve coordinated gas- and aqueous-phase chemistry. This focus on nighttime formation of secondary brown carbon is a real strength of the work. The field measurements are extensive and appear sound, largely validated in previous reports from the same campaign. The results are presented in clear figures, and the paper is well structured at the paragraph and section levels for the most part. I do strongly recommend a revision for structure at the sentence level (grammar, etc.). The scope and implications of the manuscript are suitable for ACP. I have only minor comments for the authors to consider.

**Response:** Many thanks for your comments. We really appreciated your helpful comments and suggestions. We have scrutinized the manuscript at the sentence level and revised long sentences of different sections. We also revised the language according to editorial comments of reviewer#1.

**Comment:** 180 - This paragraph includes a lot of rationale, which I usually associate with Results and Discussion, rather than Materials and Methods - should this passage be the first sub-section in the Results and Discussion?

**Response:** We agree, and moved this paragraph to Sect 3.1 to make this section more consistent with the section title "Highly absorptive SOA formed during nighttime".

**Comment:** 218 - Replace "philosophy" for clarity.

**Response:** we replaced "philosophy" with the word "framework"

**Comment:** 278 - I think the observation that the contribution of Night-OA is briefly higher than that of BBOA is overstated here - in Figure 3b, the points at hour six are almost perfectly overlapping, surely within the experimental uncertainty. Of course, the observation that the peak in the Night-OA

contribution coincides with the trough in the BBOA contribution is compelling evidence, regardless of the wording here.

**Response:** We agree, and revised this part as:

"and even approaches the contribution of BBOA near local time 06:00"

**Comment:** Figure 3 - Can error bars be incorporated into panels a, b, and d, like they are in panel c?

**Response:** We added error bars into Figure 3a. However, it would be a mess for b if error bars of all 4 lines are added without additionally necessary information to the discussion. Therefore, the error bars are not added in b. In Figure 3d, only one single value was retrieved for each wavelength, thus no error bars. We could not quantify the uncertainty arisen from the multilinear regression which is affected by many factors. The following sentence is added in the discussions:

"Note that the MAE retrieval using the multilinear regression bears uncertainties which is affected by many factors, for example, not completely independent of factors."

**Comment:** 298 - I think more detail here would be helpful, e.g., what specific values are being compared? Also, the change in wavelength dependence upon oxidative aging can be compared to trends reported in the literature, e.g., for laboratory studies of aging.

**Response:** Agree. This sentence is revised as the following:

"The retrieved $AAE_{470-590}$ for BBOA and Night OA are in general consistent with the AAE of bulk BrC solutions ($AAE_{300-600}$ of near 6) extracted using different solvents which were sampled during and after a nighttime nationwide biomass burning event (Lin et al., 2017)."

As mentioned in the text, the direct quantification of $AAE_{370-470}$, $AAE_{470-590}$ for BBOA is difficult due to the entanglement of BC absorption and thus rarely reported. We didn't find other studies that have reported AAE value changes during BBOA aging.

**Comment:** 304 - This statement implies that evaporation is the only loss mechanism for the light absorption, although photodegradation is mentioned later on line 360. I recommend introducing photodegradation here or replacing evaporation with, more generally, decay or loss.

**Response:** Thank you very much for this helpful suggestion. We agree that using evaporation might be misleading, we revised this sentence as:

"and quick daytime loss (through such as repartitioning, photodegradation, etc.)"

**Comment:** 308 - I think a few representative past studies of aqSOA should be cited here.

**Response:** Agree. Representative references were added.

"In general, SOA can either be formed through condensation of gas-phase chemically aged low- or semi-volatile VOC precursors following the gas-particle partitioning theory (Odum et al., 1996) or formed in the aqueous phase through further oxidation of water-soluble primary VOCs as well as secondary products of gas-phase VOC aging processes (Blando and Turpin, 2000;Ervens et al., 2011), with the former referred to as gasSOA and the latter referred to as aqSOA (Kuang et al., 2020a)."

**Comment:** 311 - Rephrase this sentence for clarity.

**Response:** This sentence is revised as:

"Considering the highly absorptive characteristics of secondarily formed Night-OA, exploring the sources and formation pathways of Night-OA is of great importance in meriting the importance of Night-OA formations in global atmosphere."

**Comment:** 319 - Capitalize Lagrangian.

**Response:** Revised.

**Comment:** 338 - Similarly, I think a few representative past studies of gas-particle partitioning should be cited here.

**Response:** References added.

"The gasSOA forms through condensation following partitioning theory thus adding mass mainly to condensation mode which contributes most to aerosol surface area concentrations (Kuang et al., 2020a;Zhai et al., 2023)"

**Comment:** 344 - The discussion of these two size ranges could be clarified by adding a vertical dashed line or grey band, etc., at 300 nm in Figure 4b.

**Response:** Agree. A grey area is added, and the replotted figure is as the following:

[Figure]

**Figure 1**. **(a)** Correlations between average nighttime Night-OA mass concentrations and average BBOA mass concentrations over LT16-22, colors corresponding to the average Ox ($NO_2+O_3$) mixing ratio (ppb) during the night; **(b)** Average evolutions of organic aerosol mass size distributions from local time 22:00 (20[th], 21[st] October ) to 06:00 (21[st], 22[nd] October), the grey band showing the size range of 100 nm – 300 nm; **(c)** Average differences for organic aerosol (OA) mass concentration corresponding to (b); **(d)** The correlations between increase of size-resolved organic mass concentrations in (c) (LT06 minus LT22) with average size-resolved aerosol liquid water content (ALWC).

**Comment:** Figure 5 - The x-axis in these panels is shifted from that in Figure 3. Reformatting the time scale either here or there may help readers more quickly compare between the two figures.

**Response:** Agree. To show the continuous evolution of BBOA and Night-OA as well as corresponding optical properties, we revised Figure 3 (now Figure 2 in the revised version) as the following:

[Figure]

**Figure 2. (a)** Diurnal variations of $\sigma_{BrC,370}$ and the ratio $\sigma_{BrC,370}$/BBOA; **(b)** Diurnal variations of contributions of different OA factors to $\sigma_{BrC,370}$; **(c)** Average diurnal variations of brown carbon absorption angstrom exponent between 370 nm and 470 nm; **(d)** Retrieved mass absorption efficiencies (MAE) of BBOA and Night-OA at multi-wavelengths using the multilinear regression model.

**Comment:** 391 - I recommend including a reaction scheme of this and the other gas-phase reactions

discussed throughout this paragraph.

**Response:** These reactions of $N_2O_5$ formation through $NO_3$ chemistry are common reactions in atmospheric chemistry. We added references when reactions were mentioned.

"The hydrolysis of $N_2O_5$ (Evans and Jacob, 2005;Bertram and Thornton, 2009;Wang et al., 2017;Chen et al., 2018;Wang et al., 2020) that is formed from $NO_2$ addition of $NO_3$ radicals (Seinfeld and Pandis, 2016;Fan et al., 2021), represents an important pathway of nitrate formation during nighttime period."

**Comment:** 393 - From here to the end of the paragraph, additional references to specific features in Figure 5 would help.

**Response:** Revised. References are added:

"The hydrolysis of $N_2O_5$ (Evans and Jacob, 2005;Bertram and Thornton, 2009;Wang et al., 2017;Chen et al., 2018;Wang et al., 2020) that is formed from $NO_2$ addition of $NO_3$ radicals (Seinfeld and Pandis, 2016;Fan et al., 2021), represents an important pathway of nitrate formation during nighttime period. The rapid decrease of $NO_2$ coincident with the quick nitrate formation implies that the rapid $NO_2$ consumption supplied the $NO_3$ and $N_2O_5$ reaction chains (Yang et al., 2022;Wang et al., 2023a), providing abundant $NO_3$ radical during the initial stage of Night-OA formation and likely initialized the Night-OA formation (Rollins et al., 2012;Kiendler-Scharr et al., 2016;Decker et al., 2019). The active nighttime $NO_3$ chemistry and its impacts on nitrate formations during the observation periods were further confirmed by Yang et al. (2022) who conducted box model simulations. In addition, Li, et al. (2020a) demonstrated that night $NO_3$ radical darkened the BBOA with the MAE enhancement ratio range from 1.3 to 3.2 for optical wavelength of less than 650 nm. The retrieved average MAE of BBOA through multilinear fitting was higher than the average MAE of freshly emitted BBOA during BBOA spikes as reported by Luo et al. (2022) (3.8 vs 2.6 $m^2/g$), suggesting the darkening of primary BBOA which is consistent with the prevailing nighttime $NO_3$ chemistry processes during the observations (Yang et al., 2022). The highest Night-OA production rate occurred when both $NO_2$ and NO began to increase ($O_3$ still decreased rapidly and reached below 30 ppb, shown as the blue shaded area) and the RH reached near the its maximum, which further highlights the crucial role of aerosol liquid water content in Night-OA formation. However, the quick increase of $NO_2$ and NO implies that

the dominant contribution of $NO_2$ formation to $O_3$ depletion (Wang et al., 2023a) during the quick Night-OA increase stage, and the $NO_3$ radical chemistry have ceased and likely did not directly participate in the succeeding quick aqueous-phase Night-OA formation."